# Comprehensive bathymetry and intertidal topography of the Amazon estuary

Alice César Fassoni-Andrade[1,2], Fabien Durand[1,2], Daniel Moreira[3], Alberto Azevedo[4], Valdenira Ferreira dos Santos[5], Claudia Funi[5], Alain Laraque[6]

[1]Laboratoire d'Etudes en Géophysique et Océanographie Spatiales (LEGOS), Université Toulouse, IRD, CNRS, CNES, UPS, Toulouse, France.
[2]Institute of Geosciences, University of Brasília (UnB), Campus Universitário Darcy Ribeiro, Brasília, 70910-900, Brazil.
[3]CPRM, Serviço Geológico do Brasil, Avenida Pasteur, 404, Urca, Rio de Janeiro, Brazil.
[4]Laboratório Nacional de Engenharia Civil (LNEC), Avenida do Brasil 101, Lisboa, Portugal.
[5]Instituto de Pesquisas Científicas e Tecnológicas do Estado do Amapá (IEPA), Campus IEPA Fazendinha, Macapá, Brasil
[6]IRD, GET-UMR CNRS/IRD/UPS – UMR 5562 du CNRS, UMR 234 de l'IRD, Toulouse, France

*Correspondence to*: Alice Fassoni-Andrade (alice.fassoni@gmail.com)

**Abstract.** The characterization of estuarine hydrodynamics primarily depends on the knowledge of bathymetry and topography. Here we present the first comprehensive, high-resolution dataset of the topography and bathymetry of the Amazon River estuary, the world's largest estuary. Our product is based on an innovative approach combining space-borne remote sensing data, an extensive and processed river depth dataset, and auxiliary data. Our goal with this mapping is to promote the database usage in studies that require this information, such as hydrodynamic modeling or geomorphological assessments. Our twofold approach considered 500'000 sounding points digitized from 19 nautical charts for bathymetry estimation, in conjunction with a state-of-the-art topography dataset based on remote sensing, encompassing intertidal flats, riverbanks, and adjacent floodplains. Finally, our estimate can be accessed in a unified 30 m resolution regular grid referenced to EGM08, complemented both landward and seaward with land (MERIT DEM) and ocean (GEBCO2020) topography data. Extensive validation against independent and spatially-distributed data, from an airborne LIDAR survey, from ICESat-2 altimetric satellite data, and from various in situ surveys, shows a typical vertical accuracy of 7.2 m (river bed) and 1.2 m (non-vegetated inter-tidal floodplains). The dataset is available at http://dx.doi.org/10.17632/3g6b5ynrdb.2 (Fassoni-Andrade et al., 2021).

## 1 Introduction

The Amazon River exports the largest discharge of freshwater (205'000 $m^3s^{-1}$; Callède et al., 2010) and the largest sedimentary supply (5-13 $10^8$ tons per year; Filizola et al., 2011) worldwide. However, up to now, there is not any consistent, comprehensive, publicly available topographic dataset in the estuary that can support hydrodynamic, sedimentary, or ecological studies. The largest estuary in the world is home to energetic exchanges of momentum between the upstream river and the ocean, with a marked variability of the water level over a broad range of timescales, from the semi-diurnal tide

propagating upstream from the Atlantic Ocean to the interannual hydro-meteorological climatic events frequently occurring over the upstream catchment. These exchanges between the river and the ocean result in sporadic flooding events, which profoundly impact the riparian communities' socio-economic conditions (Andrade and Szlafsztein, 2018; Mansur et al.,

2016). The morphology of the river bed is known to primarily condition the estuary's hydrodynamics, particularly the propagation of the tidal wave (Gallo and Vinzon, 2015), which is expected to affect the dynamics of the riverine floods and the extent of the associated flooding (Kosuth et al., 2009). This dynamic environment with high ecological diversity is essential for nutrient cycling and carbon fluxes (Sawakuchi et al., 2017; Ward et al., 2015), for navigation (Fernandes et al., 2007), and the transport and accumulation of sediment (Nittrouer et al., 2021).

The Amazon estuary extends from the continental shelf up to Óbidos city, corresponding to the longest tidally-influenced reach in the world extending over 800-910 km (Kosuth et al., 2009; Nittrouer et al., 2021; Fig. 1). This river flow is drained downstream towards the ocean through the main channel until the confluence with the Xingu River, around 300 km upstream of the mouth, where it is divided into two long channels, hereafter called South Channel (locally named Gurupá Channel) and North Channel (Fig. 1). Downstream of this branching, the estuary appears as a complex network of dendritic

tidal channels and islands (Fricke et al., 2019). The estuary is classified as macrotidal (Dyer, 1997; Gallo and Vinzon, 2005) and semi-diurnal (Kosuth et al., 2009) with a tidal range between 4 and 6 m at the mouth. The $M2$ (lunar semi-diurnal) and $S2$ (solar semi-diurnal) tidal constituents are the dominant components at the ocean boundary, with amplitudes of 1.5 and 0.4 m there (Gallo and Vinzon, 2005). At the upstream limit of the estuary in Óbidos, the range of the drought-flood annual cycle of the river height typically amounts to 6 m, and the tidal effects remain sensible only during the drought season

(Kosuth et al., 2009). So far, the quantitative investigation of the estuary's hydrodynamics and the interaction mechanisms between the tide and the river flow has been limited by the lack of sufficiently-resolved bathymetric databases (e.g., Gabioux et al., 2005). Past hydrodynamical studies of the Amazon estuary thus relied on approaches based on box models (Prestes et al., 2020) and/or on coarse hydrodynamical models (e.g. Gallo and Vinzon, 2015). Still, these past studies revealed rich hydrodynamics of the estuary, comprising contrasted patterns of bottom friction (Gabioux et al., 2005), active non-linear

deformation of the tidal waves (Gallo and Vinzon, 2005), a distinct structure of the salinity front (Molinas et al., 2014, 2020) and a prominent role of the intertidal flats in the flow variability (Gallo and Vinzon, 2005). The interplay between the fluvial variability of the water level and its tidal variability is particularly known to exert a central control on the estuary's sedimentation pattern (Fricke et al., 2019). While the geometry of the Amazon estuary is known to have been little influenced by anthropogenic effects up to now, it appears essential to document it in its current state, at a time when the

human influence is rising and is expected to induce profound, long-lasting impacts on the continental sediment supply to this estuary (Latrubesse et al., 2017).

The present paper aims to present a novel topography and bathymetry dataset of the whole Amazon River estuary, from its upstream limit 1000 km inland to its terminal estuary at its oceanic outlet, covering the riverbed as well as the intermittently-flooded river banks and adjoining floodplains. Over the always-wet part of the riverbed, we rely on a traditional

methodology to construct the bathymetry based on comprehensive, systematic digitization of existing nautical charts. In contrast over the intermittently-dry intertidal zones and floodplains, our mapping is achieved through an original, state-of-the-art approach based on space-borne remote sensing. Our dataset is regularly gridded at 30 m resolution and elevations are referenced to EGM08 (Pavlis et al., 2012). It covers the river streams, riverbanks, and floodplains and extends downstream of the estuary mouths over the near-shore ocean shelf and open-ocean coastline, covering the domain shown in Fig. 1.

Section 2 presents the data sources and the methods used to build the dataset. Sections 3 presents the validation against independent databases. Section 4 shows the topographic mapping and cross-section along the river and floodplain. Section 5 discusses the significance and caveats of the dataset and Section 6 explains the access to the various forms of our dataset.

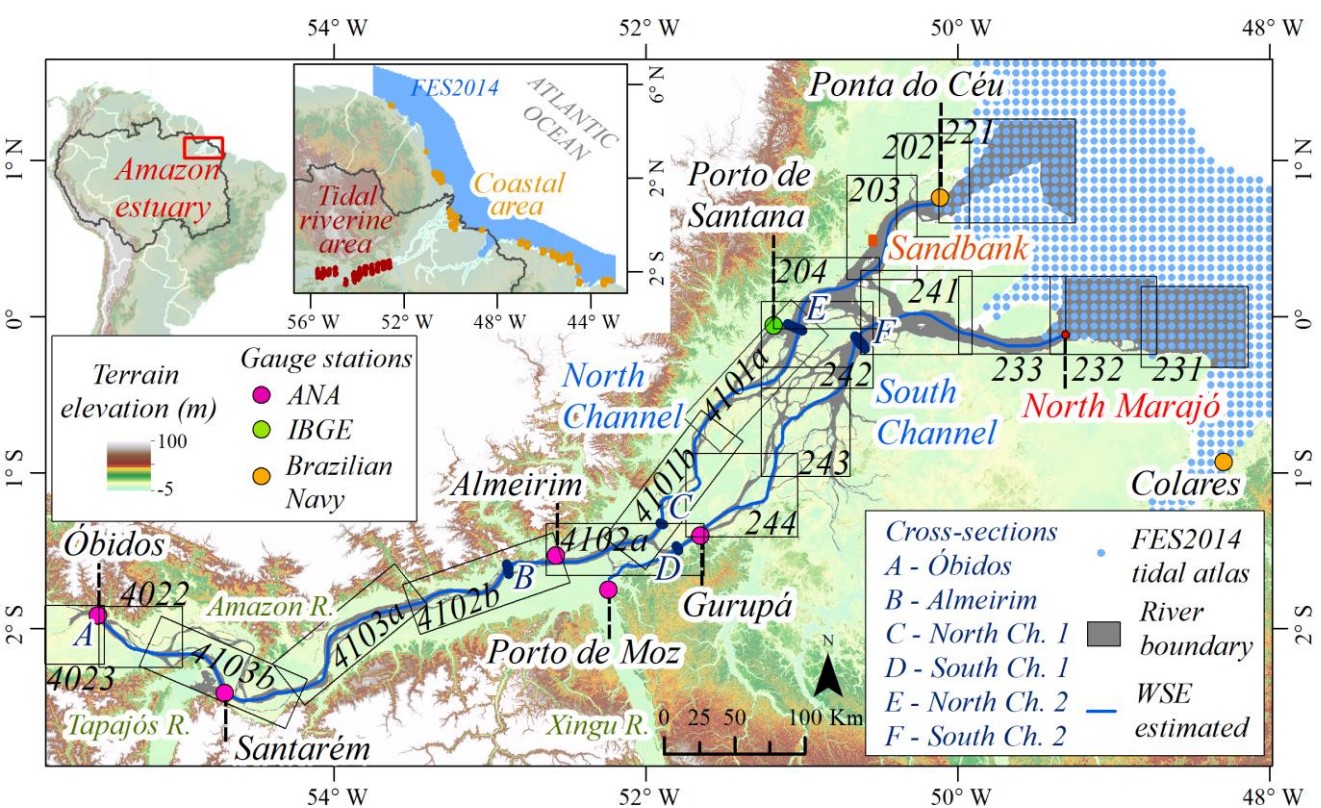

**Figure 1. Location of Amazon River estuary with identification and limits of nautical charts (Brazilian Navy - Black boxes), gauge stations from Agência Nacional de Águas (ANA), Brazilian Navy, and Instituto Brasileiro de Geografia e Estatística (IBGE). Terrain elevation from MERIT DEM.**

## 2 Data and methods

### 2.1 Bathymetry of the river bed

Over the always-wet part of the various streams of the Amazon estuary, the approach relies on systematic digitization of sounding points of bed elevation harvested from a comprehensive ensemble of nautical charts published by the Brazilian Navy (available at www.marinha.mil.br/chm/dados-do-segnav/cartas-raster). Although technically straightforward, this task was by far the most tedious part of the procedure, on account of the large geographical extent of the domain (21'500 km$^2$, a grey polygon in Fig. 1). We digitized more than 500'000 individual points in a total of 19 charts, which identification and

limits are shown in Fig. 1. The primary bathymetric surveys utilized in these charts were carried out by the Brazilian Navy on different dates varying between 1953 and 2019, with a reasonably large fraction of them done after 2000 (see Table A1 for further details). Figure 2a displays an example of a digitized nautical chart. One issue with the maps we could access was the vertical referencing of the digitized elevations. Depending on the map considered, the bed elevation values were provided with respect to two different reference water surface elevations (WSE): either the level of the 90$^{th}$ percentile of

water surface elevation (hereafter noted WS90) or the average level of the low tide during the spring tide (termed as syzygy, noted SYZ).

We inferred the vertical elevation of each of these two references from the available records of the tide gauge stations scattered along the river down to the river mouth. The tidal and limnigraphic records from the seven stations we could access, listed in Table A2 and Table A3 (locations in Fig. 1), were provided by the Agência Nacional de Águas (ANA), by

the Brazilian Navy, and by Instituto Brasileiro de Geografia e Estatística (IBGE). The vertical water level of both references (WS90 and SYZ) was deducted explicitly from the temporal records by computing the level of the 90$^{th}$ percentile to infer WS90 and by computing the average level of the low tide during spring tides for SYZ. These references were computed with respect to the geoid considering the absolute leveling published in Calmant et al. (2012) and Callède et al. (2013), complemented by a dedicated geodetic survey field that we conducted in January-February 2020 (Text A1). At the river

mouth, downstream of the downstream-most tidal stations (blue points in Fig. 1), the SYZ level was calculated using the combination of the mean sea surface height provided by the ocean general circulation model of Ruault et al. (2020) and of a proxy of the syzygy level estimated by FES2014 tidal atlas (Carrère et al., 2016; available at www.aviso.altimetry.fr/en/data/products/auxiliary-products/global-tide-fes.html). This proxy of SYZ was defined classically according to equation 1, from the sum of amplitudes of $M2$ and $S2$ tidal constituents (Pugh and Woodworth, 2014).


$$\text{SYZ} = mean\ sea\ surface\ height\ -\ (M2 + S2), \tag{1}$$

We inferred WSE (i.e., WS90 or SYZ, depending on the reference of the chart under consideration) separately along the Amazon River (Óbidos, Santarém, and Almeirim), North Channel (Porto de Santana, and Ponta do Céu) and the South

Channel (Almeirim, Porto de Moz, Gurupá, and a point of the FES2014 tidal model marked as "North Marajó" in Fig. 1) via linear interpolation between the successive stations, resulting in the profile shown in Fig. 2c. The linear interpolation considered successive points along the river spaced by 30 m and represented by the two blue lines in Fig. 1. The WSE for each of the 500'000 digitized points was then inferred from the values along the river via a nearest-neighbor interpolation method. After, the WSE was subtracted from the water depths resulting in bed elevation values referenced to EGM08. The

bed elevation points were then interpolated using the topo-to-raster method (Hutchinson, 1989), which is essentially an interpolation method suited to hydrological objects to create a regular elevation grid with 30 m spatial resolution.

In the interpolation, a river boundary was considered, as shown in Fig. 1 (grey polygon) and exemplified in Fig. 2b (bathymetry boundary). This boundary is a polygon defined considering a flood frequency comprised between 96% and 100%. The flood frequency map was calculated from the Global Surface Water (GSW) Monthly Water History v1.2 data

(Pekel et al., 2016; available at https://global-surface-water.appspot.com), which represents the space-borne Landsat-based monthly record of water presence on a global scale with a spatial resolution of 30 m. A Google-Earth engine code (Gorelick et al., 2017), described in Fassoni-Andrade et al. (2020b), was used to create it considering all GSW monthly images from the period from January 2015 to December 2018, hereby totalizing 48 months. This period of 48 months was found to be a good compromise between a short enough period to ensure the dataset is recent enough, and a long enough period capable of

capturing the bulk of the flooding statistics.

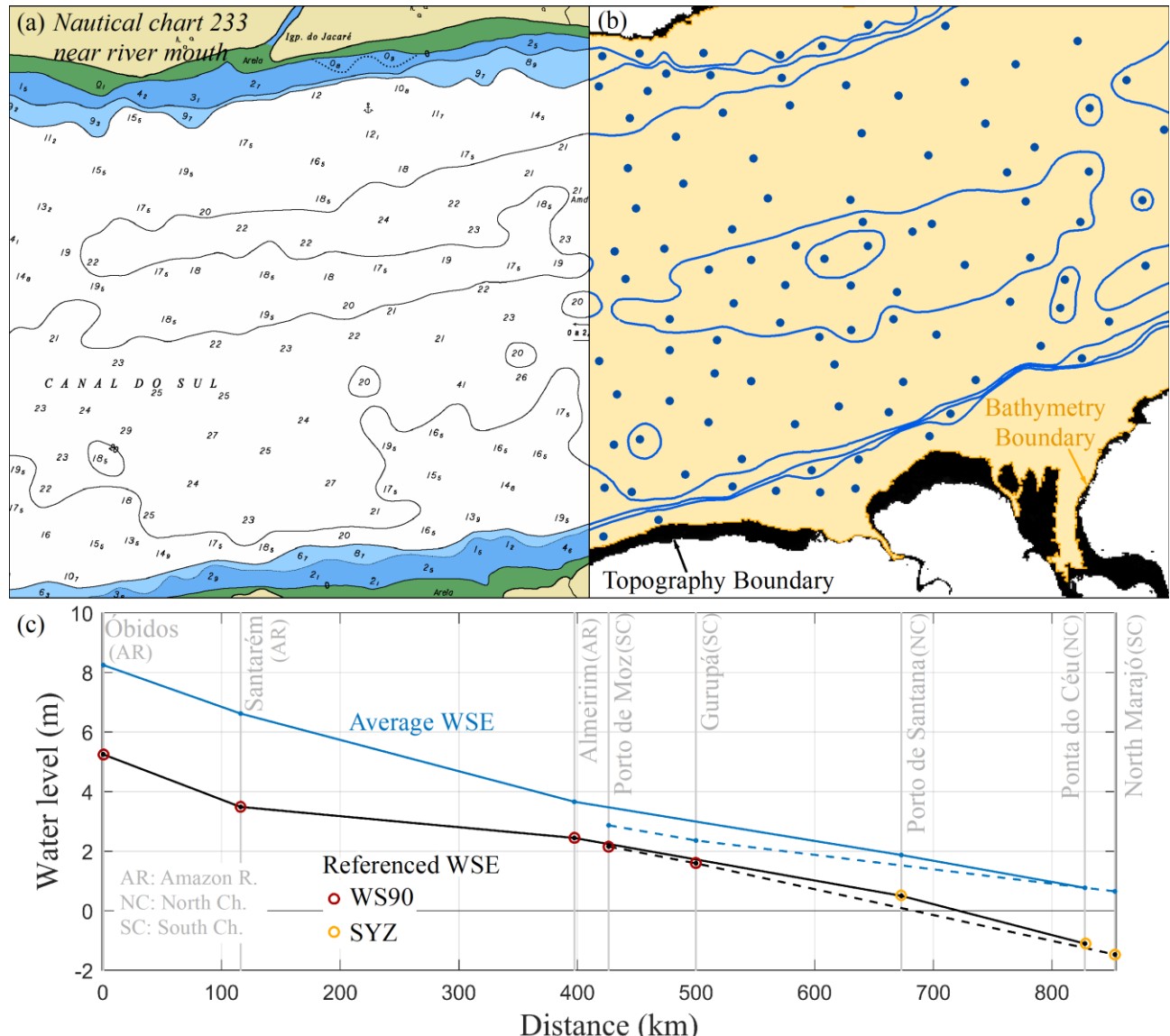

**Figure 2. (a) Example of a nautical chart nearby river mouth (code 233); (b) Digitized pointwise soundings and isobaths, along with bathymetry boundaries (defined as the isoline 96% of flood frequency) and topography boundaries (defined as the isoline 0% of flood frequency); The two regions are immediately adjacent; (c) Referenced WSE (black lines) and Average WSE (blue lines) along the Amazon estuary (EGM08 geoid).**

## 2.2 Topography of periodically flooded areas

Intertidal banks and floodplains are areas periodically flooded by tides and riverine floods, respectively. We define them as the areas comprised between 0% and 96% of the above-described flood frequency map. In the past studies devoted to coastal mapping, intertidal topography has been mapped through remote sensing data, the waterline method being one of the most widely adopted techniques (see Salameh et al., 2019 for a review). This method requires the detection and extraction of the water contours in imagery time series. Next, water levels are assigned to the individual water contours, creating isobaths. A Digital Elevation Model (DEM) raster can be generated from a large enough amount of such isobaths. Some recent applications of this method are found in Bell et al. (2016), Bergmann et al. (2018), Bishop-Taylor et al. (2019), Khan et al. (2019), and Salameh et al. (2020). This method has proven tractable with moderate-resolution space-borne imagery; however, it requires simultaneous water level knowledge at the exact time of each acquisition, along the remotely-sensed water lines. It also relies on spatial interpolation of the isobaths between successive waterlines, which can be problematic if the waterlines are sparse. Recently, alternative methods have been developed using a flood frequency map to estimate the coastal topography pixel-by-pixel (Armon et al., 2020; Dai et al., 2019; Tseng et al., 2017). In these approaches, reference water levels (for instance, minimum, average, and maximum) were assigned to reference flood frequencies (respectively 100%, 50%, and 0%). Contrasting these approaches that do not explicitly require the knowledge of the water level's temporal variation, Fassoni-Andrade et al. (2020b) related the function of water level exceedance probability and a flood frequency map to estimate the topography of the water bodies. The authors showed that the terrain elevation at a given pixel is defined as the water level, which probability of exceedance is equal to the flood frequency there. Fig. 3 exemplifies the method.

This straightforward approach requires a flood-frequency map and the water level exceedance probability functions to generate the terrain elevation map (Fig. 3). It has been applied in situations where the temporal dynamics of water filling and draining is slow. Here, the same method is applied to estimate the floodplain topography and coastal topography of the Amazon estuary, where the water level variability is spread over a broader spectrum, from intra-daily timescales to seasonal or interannual timescales (Kosuth et al., 2009). As we did not have access to vertically-leveled tide gauge archives in the Amazon estuary coastal area, two domains were considered for topographic estimation (Fig. 3). The first domain considered the riverbanks, intertidal zone, and floodplains along the channels (described in Section 2.2.1), where the approach described in Fassoni-Andrade et al. (2020a) was directly applied. Downstream of the estuary mouths, over the open near-shore Atlantic Ocean, the method was adapted to estimate tidal variation considering the water level exceedance function from a tidal station (Section 2.2.2).

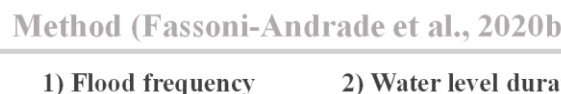

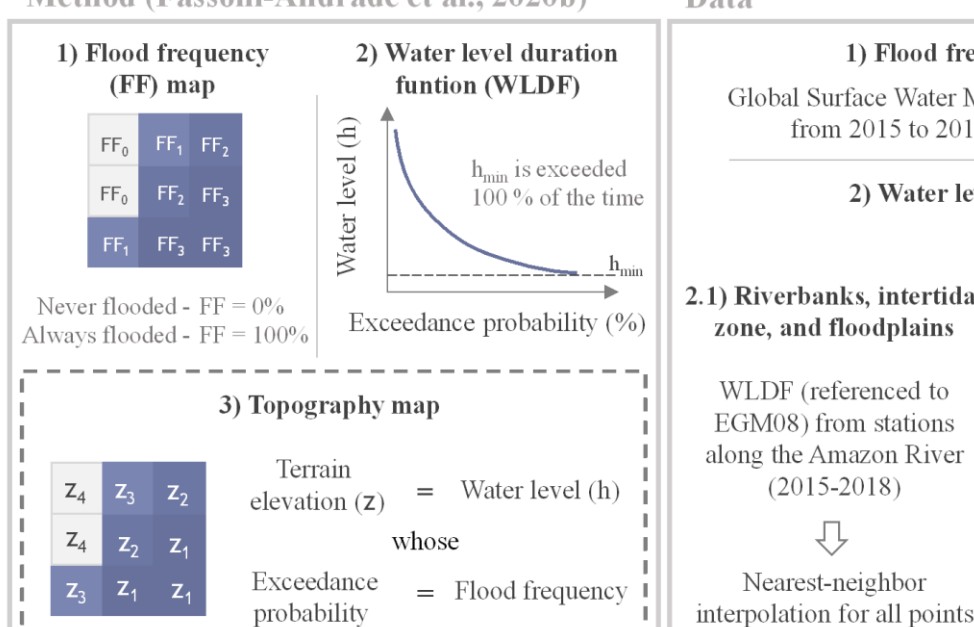

**Figure 3. Method to estimate the topography of the water bodies (Fassoni-Andrade et al., 2020b; left panel) and the various datasets considered in this study to implement this method (right panel).**

165

### 2.2.1 Riverbanks, intertidal zone, and floodplains

In the intertidal zone and floodplains along the Amazon River, WSE records from limnigraphic and tidal stations were considered over the period 2015-2018 (Table A2 and Table A3) for consistency with the imagery period covered by the flood frequency map. These records yielded exceedance probability functions, such as the one illustrated in Fig. 4b (Óbidos station). Like WSE estimation along the river (Section 2.1), the water level duration curve was inferred separately along the North Channel and the South Channel (blue lines in Fig. 1) by interpolating linearly the curves obtained at each station. Then, the water level duration curves were extrapolated through a nearest-neighbor interpolation over the whole of the estuary inter-tidal areas and floodplains, i.e., everywhere upstream estuary mouths. Therefore, the terrain elevation at any pixel was estimated considering the water level, in which the probability of exceedance is equal to the flood frequency at the same pixel (Fassoni-Andrade et al., 2020b). In permanently flooded areas, i.e., where the flood frequency is 100%, the method considers the topography equal to the lowest WSE observed, as in the river.

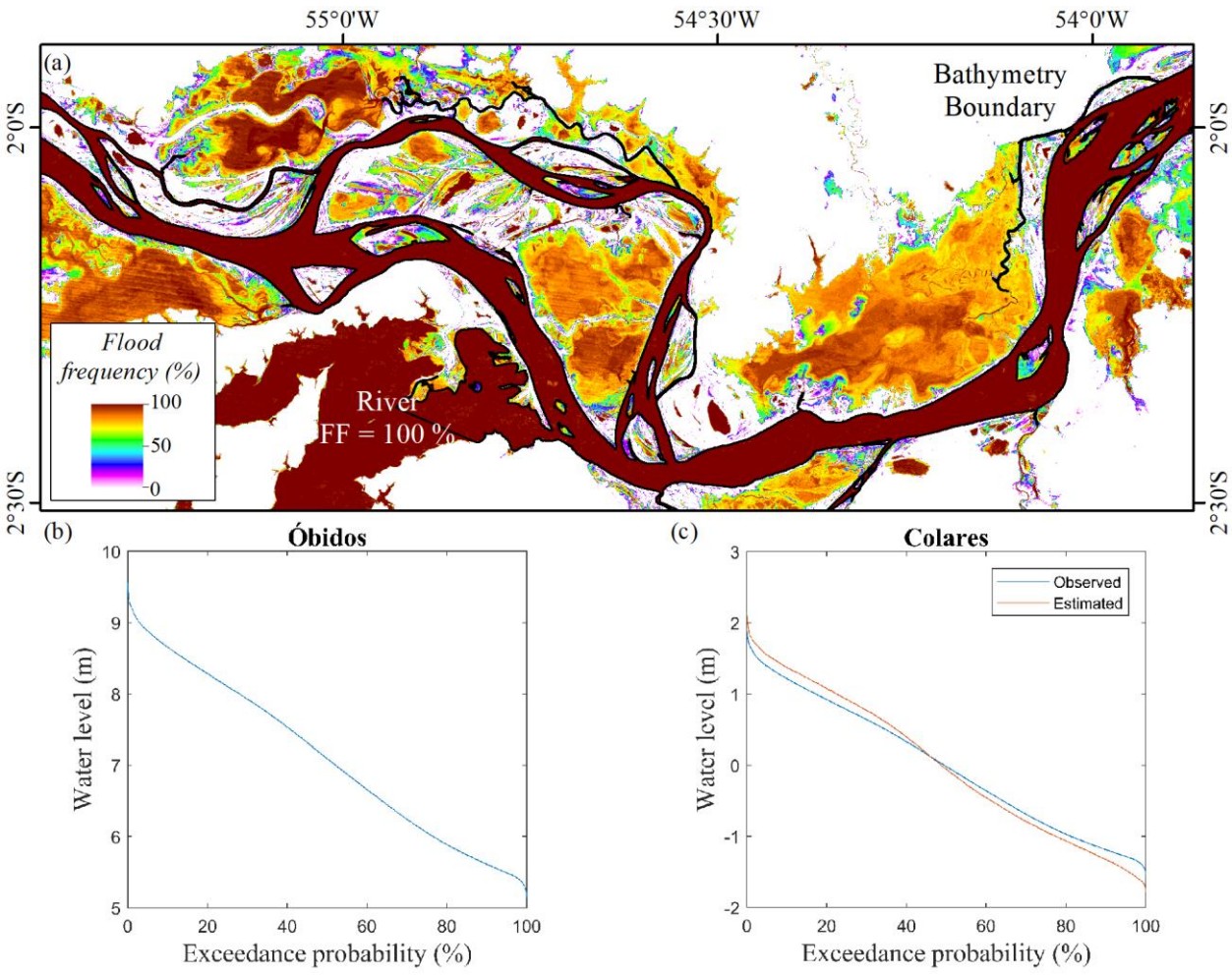

Figure 4. (a) Close-up view of the flood frequency map over the Amazon estuary's upstream part, (b) Water level duration function at Óbidos station, (c) Water level duration function observed and estimated at Colares station (see Fig. 1 for the location of these two stations).

### 2.2.2 Coastal ocean

The challenge to estimate the topography in the coastal area using the methodology of Fassoni-Andrade et al. (2020b), where vertically-leveled tide gauges are lacking, is to infer spatially-distributed water level exceedance functions. We considered the downstream-most Amazon estuary station of Ponta do Céu (Table A2; location in Fig. 1) and computed the water level exceedance function there. This curve's shape was then assumed to be the same throughout the coastal region, but with variable amplitude, proportional to the local tidal amplitude. In this semi-diurnal macro-tidal region, a reasonable proxy of the tidal amplitude over the region can be thought of as the sum of the amplitudes of $S2$ and $M2$ tidal constituents, as these two constituents are the dominant ones downstream of the river mouth (Gallo and Vinzon, 2005). The water level duration

function ($WLDF$) at any point along the oceanic coastline was obtained by scaling the corresponding function observed at Ponta do Céu ($WLDF_{PC}$) considering the tidal amplitude given by $S2$ and $M2$ components from the FES2014 model (Carrère et al., 2016), according to the equation 2:


$$WLDF = WLDF_{PC} \times \frac{\text{Tidal amplitude}}{\text{Tidal amplitude}_{PC}}$$ (2)

in which the tidal amplitude at a point is given by $2 \times (S2 + M2)$, and the PC subscript refers to Ponta do Céu values.

As a verification, Fig. 4c shows the observed and estimated exceedance probability functions at Colares station, located about 150 km to the east of the Amazon estuary (Fig. 1). Both curves represent the anomaly with respect to the average. The

two curves look very similar, with a root-mean-square deviation (RMSD) of 12.7 cm.

After estimating the water level exceedance probability at each point, the vertical reference was adjusted by matching the mean level with the height of the mean sea surface estimated by the ocean circulation model of Ruault et al. (2020). Similar to the water level exceedance probability functions estimated along the river, the coastal area's water level duration functions were inferred for all pixels of the flood frequency map considering the nearest neighbor interpolation method. Therefore, the

terrain elevation at a pixel was estimated considering the water level, in which the probability of exceedance is equal to the flood frequency at the same pixel (Fassoni-Andrade et al., 2020b).

### 2.3 In situ and space-borne data for validation

#### 2.3.1 GEA/EB Digital Terrain Model

A Digital Terrain Model (DTM) with 2.5 m of spatial resolution and accuracy of 1.62 m (BRADAR, 2017), provided by the

Instituto de Pesquisas Científicas e Tecnológicas do Estado do Amapá (IEPA; http://www.iepa.ap.gov.br/), was used to validate the estimated topography on a sandbank covering ~0.9 km$^2$ in the North Channel (location in Fig. 1). This area was chosen because it is an almost non-vegetated area and it has sufficient corresponding topographic mapping points. For consistency, the DTM vertical reference was transformed from MAPGEO2010 (Matos et al., 2012) to EGM08. The DTM was developed using P-band interferometry from an aerial survey conducted in late 2014 and early 2015 (De Castro-Filho

and Antonio Da Silva Rosa, 2017) in the context of the Base Cartográfica Continua do Amapá project (Vieira, 2015) in cooperation with Governo do Estado do Amapá and Exército Brasileiro (GEA/EB).

#### 2.3.2 ICESat-2 space-borne data

The topography was further validated against Ice, Cloud, and land Elevation Satellite-2 (ICESat-2) data. Launched in September 2018, the satellite provides measurements of the surface level from the transmission of laser pulses in the green

wavelength (532 nm) by the Advanced Topographic Laser Altimeter System (ATLAS) instrument. ATLAS beams provide

six tracks, divided into three pairs, on the Earth surface along the ICESat-2 orbit. The beam pairs are separated by ~3.3 km in the across-track direction, and each spot on the surface is ~13 m in diameter of footprint (Neuenschwander et al., 2020). The accuracy expected from ATLAS is approximately 25 cm for flat surfaces and 119 cm in the case of a 10-degree surface slope (Neuenschwander et al., 2020).

ATL08 v.3 dataset, derived from ATLAS measurements, provides every 100 m along-track heights above the WGS84 ellipsoid for the land and vegetation (available at https://nsidc.org/data/ATL08). ATL08 data can also represent water surface elevation, and therefore a criterion has been used to separate these from the measurements over the land surface. Some studies have also shown that the ATLAS instrument can penetrate water and provide information on the bottom (Ma et al., 2020; Parrish et al., 2019). Since the Amazon River has a high concentration of sediments (Martinez et al., 2009), we

assume that target information from the water represents only the water surface elevation.

Two regions were selected for the topography validation: upstream of the Xingu River, where the tidal amplitude is small (~ 40 cm; Kosuth et al., 2009), and along the oceanic coastal area (see Fig. 1 inset map for the locations of both these regions: "Tidal Riverine area" and "Coastal Area", respectively). In both regions, cloudy conditions and measurements with uncertainty above 50 cm (as indicated by the dataset flags) were discarded. Criteria to remove the points derived from the

water elevation were considered in each case. In the tidal riverine region, only ATL08 points, converted to EGM08 heights, from the October-December seasons of 2018 and 2019 were considered as this period represents the low-water season of the Amazon River. Besides, since the flooded areas should show a very low variability of the space-borne measurements along-track, it is easy to detect them in the individual along-track data. Thus, each track's water level was evaluated, and points below 50 cm above the water elevation were discarded. For the coastal area, tracks with a visually markedly different

elevation between ocean and continent were selected. Each selected track was evaluated, and points below 1 m above the water elevation were discarded.

### 2.3.3 In situ surveys of the river bed bathymetry

The river bed was evaluated in six different bathymetry cross-sections acquired from past in situ surveys done over 2007-2019 by SO HYBAM (see www.so-hybam.org) and Companhia de Pesquisa de Recursos Minerais (CPRM; Location of

cross-sections in Fig. 1). The water depth was obtained by acoustic Doppler current profiler (ADCP) instrument and a WSE was considered here for estimating the bed elevation concerning EGM08. In section A, we considered the WSE at Óbidos station on the survey day (28 November 2019). In sections, B, C, and D, the WSE at Porto de Moz station on the day of the survey was used and corrected for each section considering the water surface declivity obtained by the WSE estimated along the river (Fig. 2c) and the distance between station and section, i.e., $WSE\ at\ section = WSE\ at\ Porto\ de\ Moz +$

$WSE\ slope \times distance$. Finally, the WSE measured every 15 minutes at Porto de Santana station on 5 June of 2008 was considered in sections E and F. Since the water level varied by ~50 cm during the time span of these sections, described in Callède et al. (2010), the high-frequency WSE at each point of the sections was considered. Besides, metrics were evaluated

considering all points (excluding outliers) in the round-trip survey and repetitions. In section A, four cross-sections were acquired. In sections B, C, and D, only one cross-section was obtained. In section E, six cross-sections, and in section F, nine cross-sections were obtained.

## 2.4 Ancillary databases

The estimated topography does not cover the terrain elevation in the non-open water area. Similarly, our set of bathymetric charts does not cover the Atlantic Ocean's continental shelf downstream of the river mouth. Thus, our dataset was complemented by two global databases: Multi-Error-Removed Improved-Terrain (MERIT) DEM over the continental area and General Bathymetric Chart of the Oceans (GEBCO) v. 2020 over the ocean. MERIT DEM is a widely-used global model with a spatial resolution of 90 m in which several errors of the Shuttle Radar Topography Mission (SRTM) DEM and the height of vegetation have been corrected (available at http://hydro.iis.u-tokyo.ac.jp/~yamadai/MERIT_DEM/; Yamazaki et al., 2019). For consistency, the MERIT DEM reference was changed from EGM96 to EGM08 (Pavlis et al., 2012). GEBCO is a global terrain model referred to mean sea level with a spatial resolution of 15 arc seconds (approximately 460 m in the Amazon estuary; available at https://www.gebco.net/data_and_products/gridded_bathymetry_data/gebco_2020/). Since GEBCO data has integer values at intervals of 1 meter, the topo-to-raster interpolation was used considering the 1 m isolines to generate data consistent with float values wiping out staircases artefacts. Besides, a low-pass filter with 9 × 9 points and 19 × 19 points window moving average, i.e., 4.5 km × 4.5 km and 9.5 km × 9.5 km, respectively, was used in the region, respectively, above (shoreward) and below (off- shoreward) the -200 m isobath to reduce the noise caused by in situ multibeam sounding swaths edges.

These combined databases allowed a unified mapping of the topography and bathymetry of the Amazon estuary. However, since MERIT DEM represents the topography of 2010 and some areas in the coastal region may have been eroded or accreted between 2010 and the 2015-2017 period considered in the flood frequency mapping, a procedure was implemented to correct this issue considering three types of regions: 1) erosion areas; 2) accretion areas, i.e., regions where MERIT product does not have topographic information; and 3) GEBCO regions that represent the continent due to sparse spatial resolution, whereas it should represent transition areas or the ocean. The procedure was performed as follows: 1) MERIT DEM areas with topographic information in eroded areas were selected and replaced by GEBCO data, which covers both the continent and the ocean. In the case of substitution to continent GEBCO data, the region is corrected again in step three. 2) Deposition areas where MERIT does not have topographic information were estimated from the topo-to-raster interpolation method considering the values in the mapped regions' contours. Similarly, 3) GEBCO's high topographic regions in the ocean, including regions not previously corrected in step one, were removed, and new values were estimated by topo-to-raster interpolation considering the neighboring pixels. These areas were manually selected considering the polygons generated from the elevation reclassification into three classes: less than -8 m, between -8 m and 1 m, and greater than 1 m – visually-defined criteria. Figure A1 shows an example of these steps and a corrected area. Finally, to ensure a smooth

transition between the nautical charts and GEBCO, an area was selected and replaced considering topo-to-raster interpolation from the neighboring pixels. This area was defined by a buffer of ~2 km around the transition limit, i.e., considering 4 km width.

## 3 Validation

### 3.1 Topography

Figure 5 shows the validation of the estimated topography considering the GEA/EB DTM ("Sandbank" label in Fig. 1) and the ICESat-2 data. The estimated elevation yields an RMSE of 1.15 m, a bias of -0.78 m (standard deviation, std = 0.85 m), and a Pearson correlation coefficient ($r$) of 0.52 (number of data, n = 612) compared to GEA/EB DTM. This error may be partly related to the spatial resolution of the Landsat images (30 m) and geomorphological changes in the island during 2014 and 2015, as shown in Fig. 5ab. Still, this error is lower than the DTM intrinsic accuracy (RMSE of 1.62 m).

Considering the ICESat-2 data, the terrain elevation was also well represented in the riverbanks/floodplain and coastal area with $r$ of 0.8 and 0.8, RMSE of 1.5 m and 1.8, and a bias of 0.9 m and -1.5, respectively. However, Fig. 5f shows a bias related to the flood frequency in which overestimations are observed at low flood frequencies (e.g., ~3 m for flood-frequency of 0%). As shown in Fassoni-Andrade et al. (2020a), this bias may be related to the Landsat images used in the flood frequency map that do not represent the flood extent in flood-prone vegetated areas. Thus, the flood frequency in these areas

is considered only from situations when the water level exceeds the vegetation height, and, therefore, the flood frequency is underestimated, mechanically overestimating the terrain elevation.

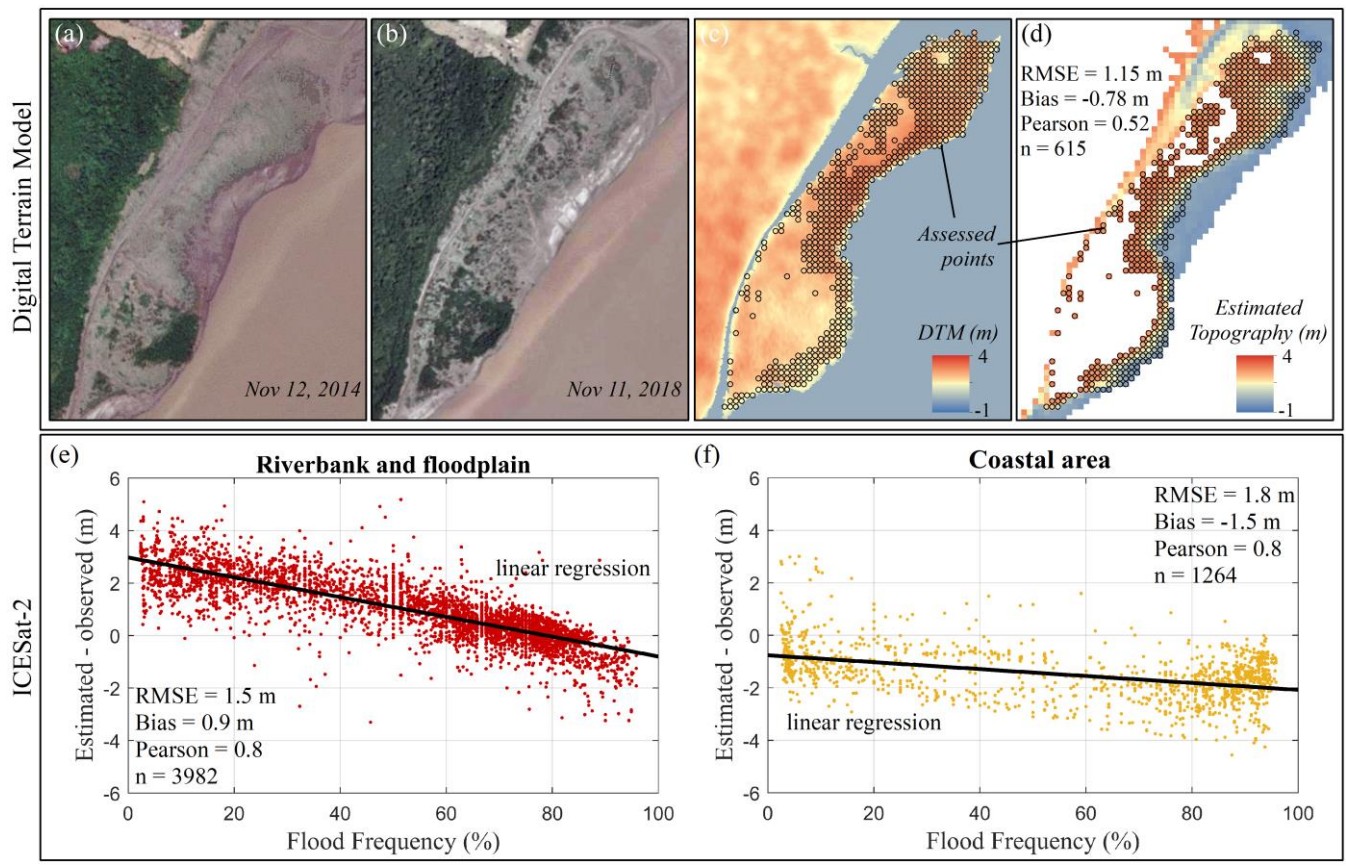

**Figure 5. Sandbank represented by GEA/EB DTM and topography mapping (location in Fig. 1). (a) and (b) represent true-color images of the area from Landsat. (c) shows GEA/EB DTM, and (d) shows the estimated topography. Flood frequency versus error in topographic estimation in riverbank/floodplain ("Tidal Riverine Area" in Fig. 1) (e), and coastal area ("Coastal Area" in Fig. 1) (f) considering the ICESat-2 data.**

## 3.2 Bathymetry

Figure 6 shows the comparison between the in situ cross-sections of the Amazon River and our product. The Amazon river's bed elevation was well represented, with an average vertical RMSE of 7.2 m and an average bias of 3.6 m (std = 5.38 m) for the six sections considered together. Keeping aside the small-scale (typically: sub-kilometric) features not resolved by the coarser bathymetric digitized charts, the shape of the cross-sections appear appropriately captured by our product, along the steep banks as well as in the median part. The smallest errors are observed in Óbidos (Section A; RMSE 3.5 m) and North Channel 2 (Section E; RMSE 2.6 m), possibly due to the shorter time lag between the dates of the surveys (< 5 years) and variation of the bed bathymetry (Vital et al., 1998). Section E, near Macapá city, is over a broad area of overconsolidated sediments, which are difficult to dredge (Vital et al., 1998). Furthermore, the WSE considered in these sections are from nearby stations, reducing the vertical reference uncertainties.

On the other hand, although the in situ survey for South Channel 2 is about 40 years old (1972-82; Section F), a good
agreement appears with the SO HYBAM survey in 2008 (RMSE of 5 m), which shows that the bed surface has possibly not
changed much in these 26-36 years.

The region near the Almeirim station, i.e., Sections B, C, and D, seems to have undergone the most significant bed change.
In general, the errors are larger (e.g., RMSE of 16 m in Section C), and the topographic variation was not represented in the
nautical charts. The respective impact of the limited spatial resolution of the digitized nautical charts and of the
morphological variability of the bed on the mismatch we observe is not clear to establish in the absence of additional
information. These are areas of intense sediment transport with erosion and/or deposition of sand and high variability in the
riverbed morphology (Vital et al., 1998). Marked seasonal changes in the riverbed were reported due to extreme net erosion,
such as the modification of a channel from a wavy bedform during rising discharge (January of 1994) to a flat floor during
low discharge (November of 1994) with the reduction of up to 7 m in the channel depth (Vital et al., 1998). Therefore, an
accurate riverbed mapping of this region remains challenging.

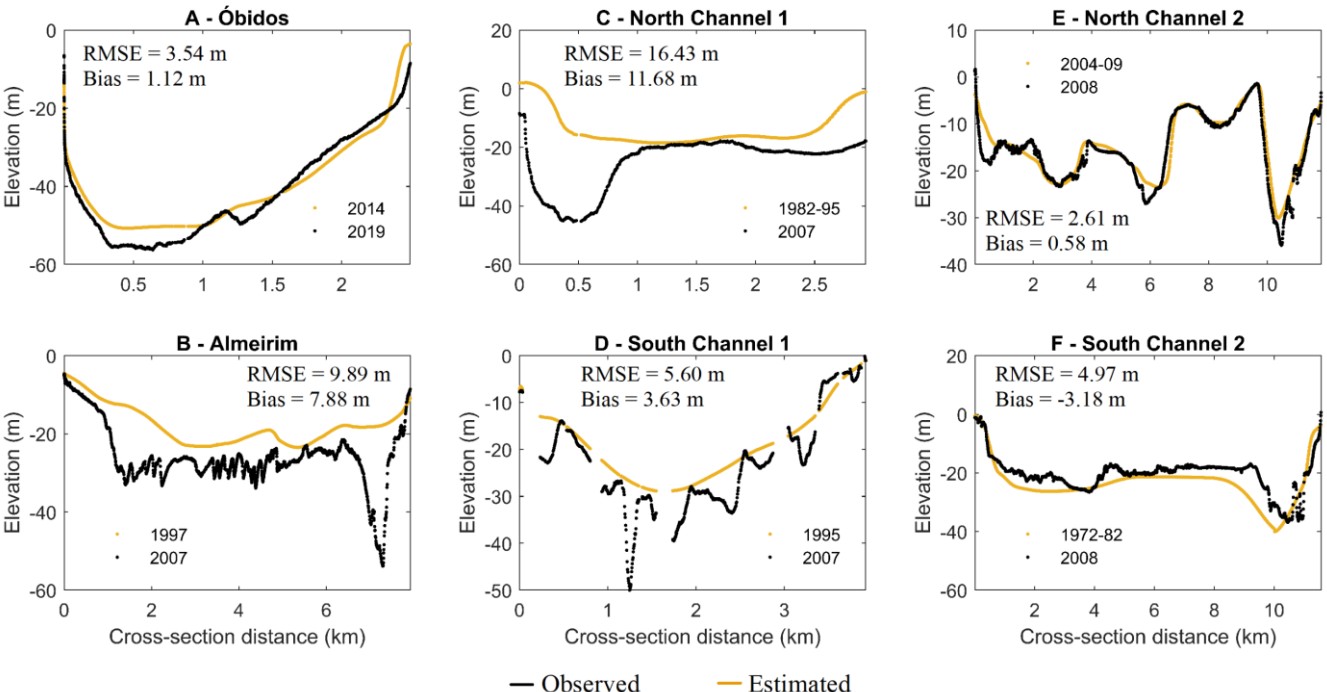

**Figure 6. Cross-section transects of the Amazon River (locations are shown in Fig. 1) from the cross-section estimated (yellow) and
observed from the in situ surveys (black). Elevations are relative to EGM08. The dates of the corresponding in situ surveys are
indicated in each panel.**

## 4 Topographic variation along the estuary

Figure 7 shows the resulting topography and bathymetry, as well as the complementary data MERIT DEM and GEBCO. The extensive floodplain, riverbanks, and intertidal zone have been seamlessly mapped, as exemplified in Box A, B, and C. It can be noted that the floodplain extent reduces from upstream to downstream, and the channels become dendritic in the eastern
half of the estuary (East of 52°W). This is related to the accumulation of sediment and the fluvial and tidal influences, as described in Fricke et al. (2019) and Nittrouer et al. (2021). These authors showed that the upper estuary, characterized by low tidal influence (~40 cm or less), has high levees that limit the overbank flow and sediment accumulation in the floodplain. In the central reach, the stronger tidal range (~1-2 m) and the associated tidal flow suppress the levees' heights, inducing strong overbank transport with high sedimentation rates in the floodplain. In the lower reach with an even stronger
tidal range (~4 m), the river canalized transport predominates, and there is little space for sediment accumulation in the floodplain.

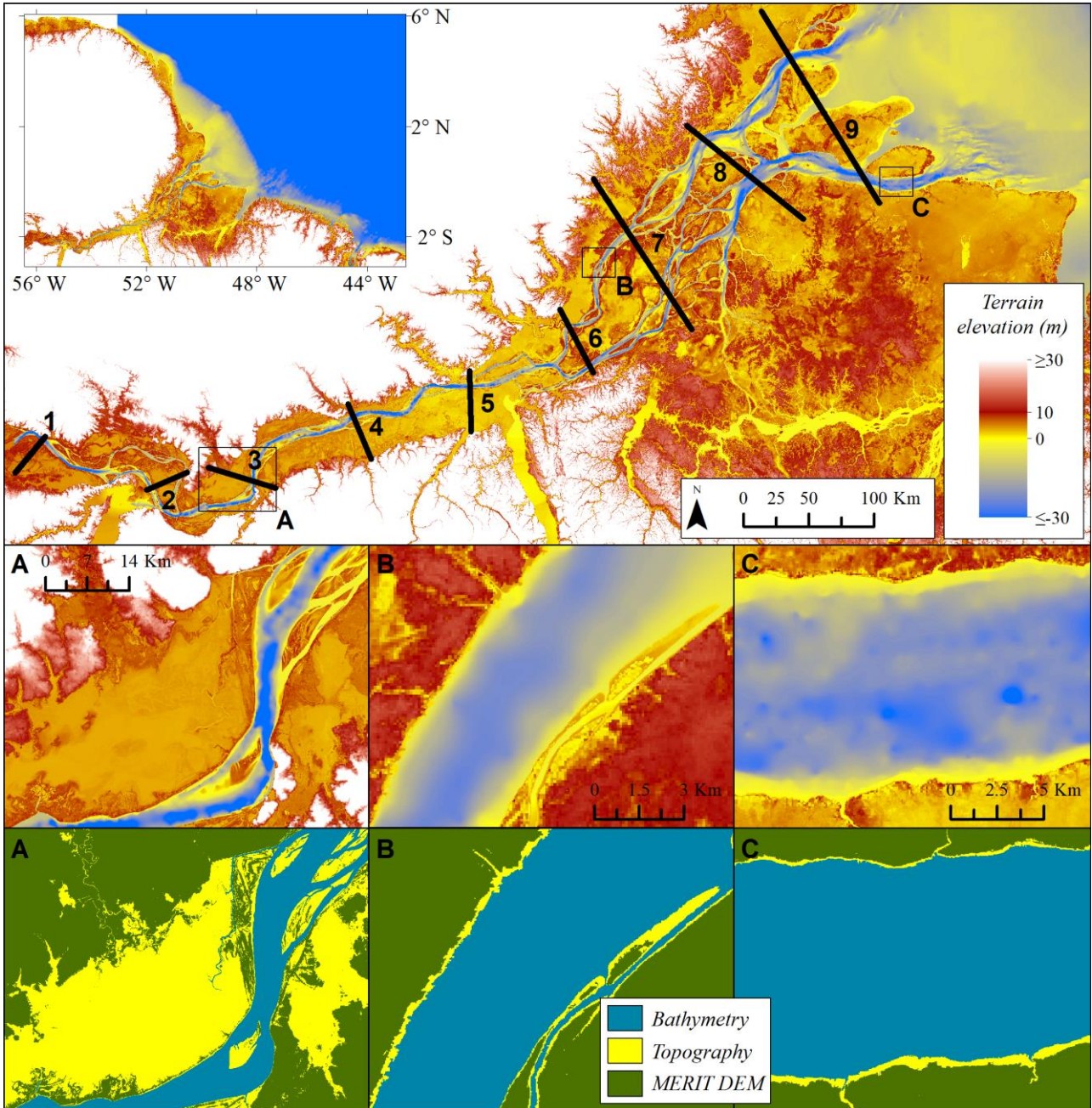

**Figure 7. Unified topographic mapping of the Amazon estuary (referenced to EGM08): merge of bathymetry, topography, MERIT, and GEBCO products. The middle row displays close-up views over selected regions. The bottom row indicates the sources of the raw data for the various sub-domains.**

In general, our product stands under this known geomorphological characterization, which is shown in the nine

representative topographic profiles of Fig. 8 (extracted in the across-river direction every 100 km). The color bar represents the flood frequency from the GSW data, i.e., regions where topography and bathymetry were estimated. Black dots represent the MERIT DEM, and the horizontal lines represent the average (blue) and the maximum (red) river WSE (2015-2018). Note the difference between the average WSE and the floodplain elevation tends to reduce from section 1 to section 5 (h values in grey); that is, the space for sediment accommodation in the floodplain reduces due to sediment accumulation. It can also be

observed that the height of the levees is similar to the maximum WSE in sections 1 to 5 (upper and central reach), but from section 6 and further downstream, the topography, represented by the black points, is higher than the maximum elevation and the river has no flooded banks. This observation in the estuary's lower reach has more uncertainty because it considers the MERIT DEM data (Yamazaki et al., 2019), which were not validated here. Fricke et al. (2019) did not observe levees in this reach and described the topography as a flat surface, but the evaluation of the authors considered topography surveys of

the banks with a distance from the river of 30 to 250 m (average of 80 m), which is equivalent to ~3 points of our analysis (30 m of spatial resolution).

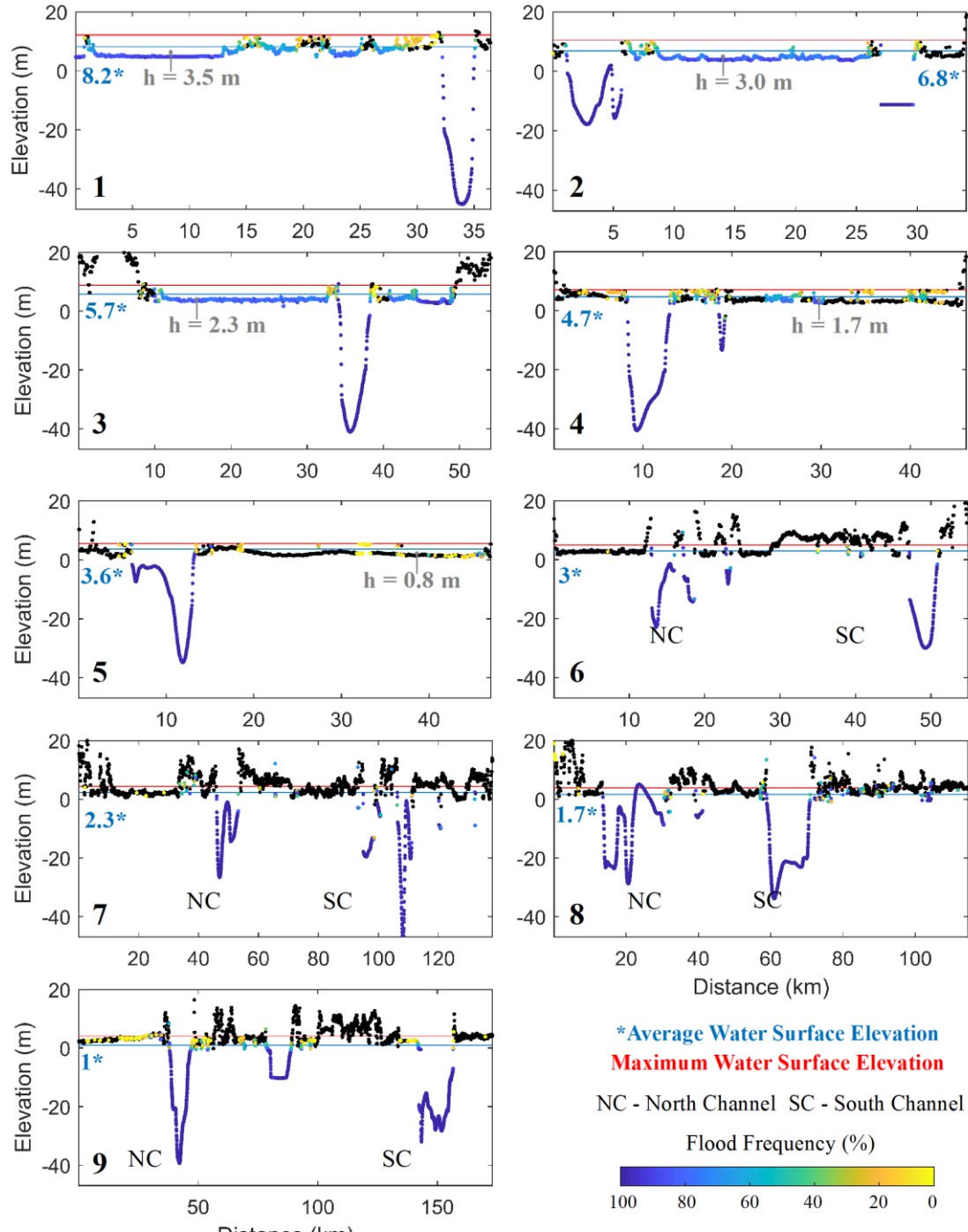

**Figure 8. Topographic profiles distributed every 100 km along the river, with the flood frequency, average and maximum WSE,**
**and height between the average level and the minimum elevation of the floodplain.**

# 5 Conclusion

## 5.1 Summary and significance of the dataset

Our dataset provides the first-ever consistent, high-resolution, and vertically-referenced topography of the Amazon estuary. Our product's vertical accuracy typically amounts to 7.2 m (bias of 3.6 m) over the river bed and 1.2 m over non-vegetated inter-tidal floodplains (2015-2018 period). These values appear in line with similar remote, poorly-surveyed tropical or deltaic shorelines (Khan et al., 2019; Salameh et al., 2019). Our mapping is based on an innovative approach using remote sensing data, an extensive and novel dataset of river depth, and auxiliary data over the adjoining areas. We believe that this new approach opens unprecedented opportunities for a straightforward estimation of coastal topography worldwide. The validation approach uses an independent and spatially-distributed dataset of various origins (in situ and remotely-sensed), which provides vital support to our findings' quality.

Our overarching goal in assembling this dataset is to characterize the topography and bathymetry of the world's largest estuary. This dataset has many potential applications, such as hydrodynamic modeling, flooding hazard assessment, sedimentology, ecology, physical or human geography, among others. For hydrodynamic modeling, for instance, where the knowledge of topography is instrumental for the accuracy of the results, as well as for geomorphological assessments, which are usually performed with satellite images extracting horizontal information (e.g., width, length) but most often lacking the vertical information, we believe this dataset offers a substantial potential of scientific progress. The dataset can also support ecological studies such as vegetation distribution and carbon balance.

The availability of high-resolution space-borne imagery promised by ongoing operational initiatives (such as the European program Sentinel or the upcoming Constellation Optique 3D - CO3D - mission) opens excellent prospects for frequent revisit updates and improvement of the intertidal part of our product. Keeping in mind the Amazon estuary's energetic morphodynamics, such updates will ensure a perennial quality of our dataset.

## 5.2 Caveats

Our product's main limitation lies in the long timespan of our raw bathymetry data collection (encompassing five decades, Table A1). This limitation is probably sensible regarding the supposed characteristic timescale of the variability of river bed through erosion and accretion processes, as revealed from our validation. Repeated shipborne bathymetric surveys are needed, although the geographical extent of the domain makes it hardly tractable at this mega-delta scale. In particular, it will be opportune to consider the future releases of bathymetric charts by the Brazilian Navy along the Amazon estuary, as they become available in the future years, in case they are based on updated primary bathymetric surveys. The issue is less severe for the intertidal topography, as the timespan of our primary data period is inferior by one order of magnitude (4 years only). Inherently, our product relies on GEBCO digital terrain model in the open-ocean region. As such, we are subject to the same sources of error as everywhere else in the world ocean, related to the poor knowledge of the vertical datum of some of

the primary data used in GEBCO composite product (Weatherall et al., 2015). Our product is also potentially impacted by the inhomogeneity of the quality of GEBCO digital terrain model, in particular in the near-shore oceanic regions (Amante and Eakins, 2016).

Another limitation of our dataset over the intertidal flats and floodplains results from the approach based on remotely-sensed imagery of GSW product to estimate the flood frequency. Indeed, it is not expected to work well over the Amazon estuary areas that are densely vegetated. In addition, topographic mapping bias due to flooded vegetation could be avoided using satellite radar data to map the water extent even in flooded vegetation, such as ALOS/PALSAR (Arnesen et al., 2013). Another issue of using Landsat images for coastal topography estimation is the flood extent representation only every 16

days (Landsat has a sun-synchronous orbit). The tide's temporal variability occurs on an hourly scale, and the amplitude of $S2$ tidal constituent would be observed in the same phase, introducing a bias in the mapping. More investigations are needed using images with more significant temporal variability. The upcoming SWOT satellite mission that will provide for the first time a frequent mapping of the water surface elevation and water extent over continental and riverine areas offers a bright prospect to curb these limitations.

**6 Data availability**

The dataset generated from this work (listed below) is available at http://dx.doi.org/10.17632/3g6b5ynrdb.2 (Fassoni-Andrade et al., 2021). All other datasets used in our work are open-source data cited within.

1) Bathymetry of the Amazon estuary (Bathymetry.tif and Bathymetry.nc); Elevation in meters relative to the EGM2008 geoid.

2) Topography of the non-forested portion of the lower Amazon floodplain (Topography.tif and Topography.nc); Elevation in meters relative to the EGM2008 geoid.

3) Flood frequency for the period 2015 to 2018 (FloodFrequency_15to18.tif and FloodFrequency_15to18.nc); Values ranging from 0 to 100%.

4) Unified mapping of the Amazon estuary: merge of bathymetry, topography, MERIT, and GEBCO products

(DEM_AMestuary.tif and DEM_AMestuary.nc); Elevation in meters relative to the EGM2008 geoid.

5) Boundaries of each domain of the unified mapping (domain_boundary.shp).

6) Code example to read netcdf files in MATLAB (read_netcdf.m).

**Appendix A**

**Table A1. Identification of nautical charts and dates of surveys (Brazilian Navy)**

| Nautical chart | Dates |
| --- | --- |
| 4023 | 2013-2016 |

| 4022 | 1986/2013-2014 |
| 4103b | 1990/1998/2003-2007/2011-2014 |
| 4103a | 1998/2007/2009 |
| 4102b | 1978/1997/2012 |
| 4102a | 1978/1995/1997 |
| 4101b | 1969-1975/1982-1995/2005 |
| 4101a | 1969-1978/1991-1993/2004-2009/2011-2012 |
| 244 | no information |
| 243 | no information |
| 242 | 1972-1982/1983-1986/1991-1993/2004-2012 |
| 241 | no information |
| 233 | no information |
| 232 | 1973/2004 |
| 231 | no information |
| 204 | 1972/1983-1993/2004-2009/2011 |
| 203 | 1977/1980 |
| 202 | 1953-1956/1980/1989-1991/2017/2019 |
| 221 | 1970/1994-1989/2005-2008/2017-2019 |


**Table A2. Gauge station in the North Channel of the Amazon estuary**

| | **Óbidos** (-55.51, -1.92) | **Santarém** (-54.70, -2.42) | **Almeirim** (-52.58, -1.53) | **Porto de Santana** (-51.18, -0.06) | **Ponta do Céu** (-50.11, 0.76) |
|---|---|---|---|---|---|
| **Source ID** | ANA/CPRM 17050001 | ANA/CPRM 17900000 | ANA/CPRM 18390000 | IBGE | Marinha 10653 |
| **Frequency** | Daily | Daily | 15 minutes | 5 minutes | 10 minutes |
| **Absolute vertical correction (cm; EGM08)** | 358*** | 190.6*** | -39 (FS) | -- | -- |
| **Relative vertical correction (Eq) (cm; EGM08)** | -- | -- | -- | AL -187* | AL – 77** |
| **Period (Topography)** | 2015-2018 | 2015-2018 | 2017 a 2018 | 2015-2017 | 11 Feb 2008 3 Nov 2008 |
| **Period (Bathymetry)** | 22 Feb 1968 30 Nov 2019 | 01 Sep 1930 31 Oct 2019 | 12 Mar 2015 30 Sep 2019 | 14 Apr 2016 30 Apr 2018 | 11 Feb 2008 3 Nov 2008 |

| Reference WSE (cm; EGM08) | WS90 525 | WS90 348.6 | WS90 244.4 | SYZ (n =57) 50.04 | SYZ (n =17) -110.38 |
| --- | --- | --- | --- | --- | --- |

FS – Field Survey (Text A1).

Eq = Average level (AL) - level above the geoid (EGM08).

*(Callède et al., 2013); **(Ruault et al., 2020); ***(Calmant et al., 2013).

**Table A3. Gauge station in the South Channel of the Amazon estuary**

| | Porto de Moz (-52.24, -1.75) | Gurupá (-51.65, -1.41) | North Marajó (-49.37, -0.18) |
| --- | --- | --- | --- |
| **Source ID** | ANA/CPRM 18950003 | Kosuth et al. 2009 | FES2014 |
| **Frequency** | 15 minutes | 30 minutes | -- |
| **Absolute vertical correction (cm; EGM08)** | 39.7 (FS) | -- | -- |
| **Relative vertical correction (Eq) (cm; EGM08)** | -- | AL - 236* | AL – 65** |
| **Period (Topography)** | 2015-2017 | 24 Jan 2000 21 Oct 2000 (***) | -- |
| **Period (Bathymetry)** | 27 Oct 2014 28 Jan 2020 | 24 Jan 2000 21 Oct 2000 (***) | -- |
| **Reference WSE (cm; EGM08)** | WS90 215.7 | WS90 159.16 | SYZ ($M2 + S2$) -147 |

FS – Field Survey (Text A1).

Eq = Average level (AL) - level above the geoid (EGM08).

*(Callède et al., 2013); **(Ruault et al., 2020).

*** data from October 2000 are repeated twice to complete one year of data.

**Text A1.**

Series of stage values are relative to a so-called "gauge zero", which simply corresponds to the lowest mark on the graduated staff and is referred to an arbitrary datum that is different in each gauge station. Therefore, stages from one gauge cannot be compared in an absolute way to stages at other gauges. It is not possible to obtain the corresponding water surface elevation to derive, for example, the slope of the water surface or relate the water level to a Digital Elevation Models-DEM of the surrounding watershed. Yet, the slope information is a key-parameter for the hydrodynamic modeling of the flow in the basin. The "zero values" of the Almeirim and Porto de Moz gauge stations were surveyed using GNSS (Global Navigation Satellite System) geodetic receivers installed over gauges benchmarks. The data surveyed was computed in Precise Point

Positioning (PPP) technique (Héroux and Kouba, 1995), using the GINS software (Marty et al., 2011) developed by the French Space Agency (CNES). Coordinates were produced in the WGS84 ellipsoid related with ITRF2014 frame and following all the recommended corrections from the IERS 2010 conventions (McCarthy and Petit, 2011). The efficiency and accuracy of GINS to process GPS data in PPP mode expected from our processing chain is better than 2 cm. This expected accuracy is possible thanks to the GNSS observation time, and the model corrections accuracy (see Moreira et al., 2016 for

further details).

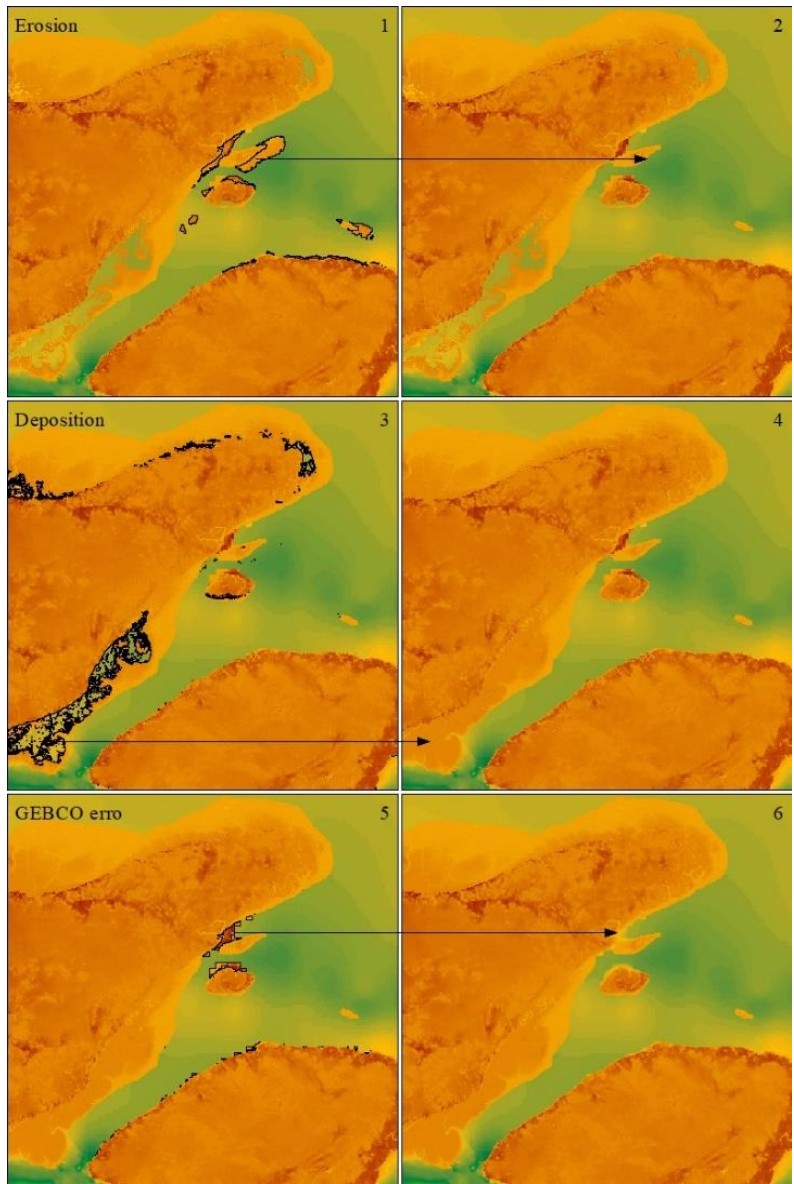

**Figure A1. Example of data correction in ocean after merging the database (GEBCO, MERIT DEM, bathymetry and topography).**

## Author contributions.

AF, FD: Conceptualization, Data acquisition, Formal analysis, Methodology, Validation, Writing - review & editing; DM: Data acquisition, Methodology, Validation, Writing - review & editing; AA: Methodology, Validation, Writing - review & editing, Project administration; VF, CF, AL: Data acquisition, Writing - review & editing.

**Competing interests**

The authors declare that they have no conflict of interest.

**Acknowledgements**

EOSC-SYNERGY (grant number 857647), IRD, CPRM, LAGEQ/IG/UnB for financial support. Leandro Guedes Santos (CPRM-Belém), Rodrigo Da Silva (UFOPA - Santarém), Fabrice Papa (IRD-LEGOS), Victor Hugo da Motta Paca (CPRM-Belém), Arthur Abreu (CPRM-Rio de Janeiro) and the whole crew of R/V Isabella for logistical support during the

"Dinâmica Fluvial 2020" bathymetric cruise. Gérard Cochonneau (from SO HYBAM-IRD) for sharing the in situ tidal records published by Kosuth et al. (2009). Julien Jouanno (IRD-LEGOS) for sharing the mean sea surface of the ocean circulation model of Ruault et al. (2020).

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
