# Peer review of "Comprehensive bathymetry and intertidal topography of the Amazon estuary"

_Earth System Science Data, 2021_

## Referee Comment (RC2)

**General comments**

As reviewer, I recognize the quality of the work undertaken to realize a topo-bathymetric DEM in such a complex environment. I am particularly attentive to actions related to the tedious work of digitizing charts, plus all the detailed work done to properly take into account the different vertical datums and above all the innovative way of getting elevations from the flood exceedance method. The reviewer appreciate that you illustrate the inter-relation between these datums as it brings forward to the mind of the reader this issue in terms of vertical accuracy of your data product.

**Specific comments**

Line 113:As many planar interpolation tools are existing to grid point cloud data, could you please further detail why you have selected the topo-to-raster algorithm (general principle, pros and cons of this algorithm).

Section 2.2: We suggest a schema to better describe your methodology and make it easy for the reader to understand it.

Figure 3.a I believe the color ramp is not fully appropriated to what you want to describe. Indeed you are using a divergent color map centered close to 50%. It would be preferable to use a sequentially progressing color map (like blue to red, or light yellow to brown, without going through white). Few comments on the analysis of this map: I do not understand the strong boundary between dark blue and yellow (SSW of the map). Also, it is my understanding that flood exceedance should be high close to the border of the river (banks, coast) and small far away from it (where the altitude is higher). It does appear to be the opposite from your map. Please comment or correct. If this a poor interpretation from me, please help me (and other readers) not get confused and/or better understand your methodology and how to interpret this map.

Section 2.3.3. This section seems to indicate that a 2D representation of the WSE would be welcome.

GEBCO: GEBCO is mainly originating from Hydrographic Offices data. These organizations tend to provide bathymetric data relative to local "Chart Datum" which can be roughly estimated to be equivalent to the Lowest Astronomical Tide. While in deep ocean (main objective of this DTM) Lowest Astronomical Tide (LAT) and Mean Sea Level (MSL) can be safely assimilated to be convergent, it is not the case in coastal waters. To my knowledge, no vertical shifting consideration from CD to MSL has been taken into account in the overall GEBCO production. This should be considered as a serious limitation having major consequences on the vertical precision of the bathymetric DTM.

Another important point to consider with GEBCO is that it is a composite DEM with multiple sources of different intrinsic horizontal and vertical resolution. Locally, using the TID (accompanying grid) we can see that bathymetric information in the GEBCO grid are originating from "interpolated based on a computer algorithm" but we do not have any more details on the underlying data and their characteristics (density, vertical and horizontal accuracy). Hence, it is not fully advised to rely on the transition between land and sea from GEBCO.

[Figure]

Figure 4: Here again I believe the color-ramp is not adequately chosen (gray at 4m can be easily confused with cyan at -1m) prefer here also a sequential color-ramp.

Figure 6. Please indicate the vertical reference in the legend. Also note that as Bed elevation and Terrain elevation should have a complementary color bar while they seem to intersect between 0 and -1m

Conclusion/Caveat: I would have preferred a more detailed discussion section rather than a "caveat section" in the Conclusion. However, I recognize the rigorous and "ethical" state of mind of the authors in stating the limitations of their data product.

May I also suggest to the authors to:

- undertake a simple slope or shading (in different orientation) which may highlight some artefacts or poor discontinuities in their model

- provide a mask indicating where the is the limit of your DEM, with the level of accuracy you describe in your paper. Moreover, I would suggest providing an associated grid describing the origin of each nodes of your DEM (like the TID concept for the GEBCO grid).

May I also suggest the following reading:

[1] P. Weatherall et al., "A new digital bathymetric model of the world's oceans," Earth Sp. Sci., vol. 2, no. 8, pp. 331–345, Aug. 2015.

[1]     C. J. Amante and B. W. Eakins, "Accuracy of Interpolated Bathymetry in Digital Elevation Models," *J. Coast. Res.*, vol. 76, pp. 123–133, 2016.

[1]     C. J. Amante and B. W. Eakins, "Accuracy of Interpolated Bathymetry in Digital Elevation Models," *J. Coast. Res.*, vol. 76, pp. 123–133, 2016.

---

## Referee Comment (RC3)

**Comprehensive bathymetry and intertidal topography of the Amazon estuary**

Alice César Fassoni-Andrade[1], Fabien Durand[1,2], Daniel Moreira[3], Alberto Azevedo[4], Valdenira Ferreira dos Santos[5], Claudia Funi[5], Alain Laraque[6]

[1]Institut de Recherche pour le Développement (IRD), Universidade de Brasília (UnB), Brasília, 70910-900, Brazil.
[2]LEGOS UMR5566, CNRS/CNES/IRD/UPS, 31400 Toulouse, France

[revised manuscript text omitted]

---

## Author Response (AR1)

**Reviewer#1**

**General comments**

**The paper titled "Comprehensive bathymetry and intertidal topography of the Amazon estuary" by Alice César Fassoni-Andrade et al., presents a wide overview of a much-needed topographic-bathymetric data set of the Amazon Estuary area. It is composed of a combination of existing maps (topography and ocean bathymetry) with a newly developed dataset, based on a novel approach to map the topography using flood frequency. This new dataset can be used in hydrodynamical modelling and in other applications that need such a high-resolution data. The authors clearly presents the dataset errors and caveats, which is highly appreciated, and provide the data for future use. I therefore think the paper (and the data) is suited for publication after addressing a few issues detailed below. It will be clear from the points I raised that I am not an expert in this kind of environment or datasets, yet still, there are a few points that needs to be addressed by the authors.**

Reply: We are thankful for your feedback and for your detailed review.

**Specific comments:**

**Regarding the online dataset – I had some problems opening and using the netcdf version of the maps. Can you add a short explanation on how to use them? The geoTIF files were working fine. Additionally, I think it could be very helpful if you add a meta-data map to the dataset. I.e. a map containing the spatial distribution of the different data sources, and the areas that were smoothed etc.**

Reply: The netcdf format has been a standard format in large oceanographic and meteorological datasets for the past 20 years. The distinct advantage of this format is that it is self-explanatory, in that all the required metadata are embedded in the files, under the form of attributes, so that one does not need to provide separate metadata files for the final user to understand the file content.

We agree it is useful to provide a short explanation about how to use the netcdf files. We incorporated a matlab file in our Mendeley repository (http://dx.doi.org/10.17632/3g6b5ynrdb.2) with a code example to read netcdf files (read_netcdf.m). We have also included a shapefile with the boundaries of each domain of our DEM (domain_boundary.shp), which distinguishes 5 categories 1) Bathymetry (with nautical charts as data source); 2) Topography of periodically flooded (with our original spaceborne dataset); 3) MERIT DEM; 4) GEBCO v2020 and 5) Interpolated areas. This new information is referred to at line 433 of the revised manuscript.

**L122: Please explain why you only 4 years of data for the flood frequency. If I remember correctly, Pekel's dataset comprises >30 years of data. This could be a major drawback of your dataset, since 4 years is not so representative if accounting for hydroclimatological events.**

Reply: The goal of selecting 4 years is to comply with our assumption of stationarity of the dataset. Indeed, the Amazon banks topography is known to vary significantly at interannual-to-interdecadal timescales (see e.g. Gensac et al. CSR 2016, http://dx.doi.org/10.1016/j.csr.2016.02.009). Hence it is desirable to select a time window as short as possible in Pekel's dataset, to be sure to capture the most recent state of the topography. On the other hand, our methodological approach requires the dataset to capture the full statistics of the water level, including the low and high extremes. Keeping in mind that the water level variability over our domain is primarily dominated by the annual periodicity of the flood-drought hydrological cycle (Kosuth et al., 2009), one could even think of selecting only one year of Pekel's data. However, given that the region is characterized by a

fairly high cloudiness (Martins et al., 2018), it was found that a 4 year long window was a good compromise between the quality and the quantity of primary data in Pekel's product. This particular 2015-2018 period has the advantage of comprising an excess flood year (2015), a drought year (2016), and more normal years (2017, 2018). To make it clear to the reader, we added the following sentence (lines 126-128):
"*This period of 48 months was found to be a good compromise between a short enough period to ensure the dataset is recent enough, and a long enough period capable of capturing the bulk of the flooding statistics*".

**L215: Did you convert the ellipsoid heights to EGM08 heights?**
Reply: Yes we did. We have included this information on line 239.

**L220: Was this assumption validated somehow? Even visually?**
Reply: This assumption is indeed hard to validate, in the absence of systematic data of the water turbidity in the Amazon, to the best of our knowledge. Still, we believe it is a reasonable statement, as Parrish et al. (2019; reference cited in the preceding sentence of the manuscript) concluded that ATLAS instrument is basically blind beyond 1 Secchi of water depth, which typically amounts to less than 30cm in the Amazon (Nakajima et al. 2017, doi: 10.7717/peerj.3308).

**L250: How did you correct from EGM96 to EGM08?**
Reply: we did so by simply adding the spatially-varying difference between the two models.

**L255-258: Can this explain the trend in error you show in Fig 4e & 4f? It seems like there is a systematic trend in this error, which to my opinion is strange. Shouldn't this bias be corrected?**
Reply: The error in the GEBCO data is not the cause in the trend since the GEBCO data has not been validated. Figure 4ef (now Figure 5ef in the revised version of the manuscript) shows a trend of higher (lower) error at low (high) flood frequencies. This bias may be related to the error of the inundation extent mapping in the Landsat optical images (Pekel's data). Thus, inundated vegetation areas are considered inundated only when the water level is higher than the vegetation height, causing a bias. In previous study (Fassoni-Andrade 2020; Fig. 8), we have shown that this error is related to vegetation height indeed. We chose not to correct the bias but included this limitation in the text (lines 413-417).

Reference : Fassoni-Andrade, A. C., Paiva, R. C. D., Rudorff, C. M., Barbosa, C. C. F. and Novo, E. M. L. de M.: High-resolution mapping of floodplain topography from space: A case study in the Amazon, Remote Sens. Environ., 251, 112065, doi:10.1016/j.rse.2020.112065, 2020.

**L295 and sect. 2.3.3: Why did you choose these specific cross sections? Were these the only one existing? If you add other cross sections to the validation – will the resultant error change significantly?**
Reply: Yes unfortunately we acknowledge that these six cross-sections are the only ones available to us. Given that they are well spread along the course of the estuary, from its upstream limit to its mouths, we are hopeful that our error estimation is somehow robust. But we are alas unable to quantify this.

 **Technical corrections**

**L15: "this characterization" – what characterization? Do you mean this "mapping"/"product"?**
Reply: Yes me meant "mapping". We changed this sentence accordingly (line 15).

**L26-27: "the largest"… of what? Do you mean the largest of all rivers? If so, please state this.**
Reply: We agree. We modified the sentence accordingly (lines 26-27):
"*The Amazon River exports the largest discharge of freshwater (205'000 m3s-1; Callède et al., 2010) and the largest sedimentary supply (5-13 108 tons per year; Filizola et al., 2011) worldwide.*"

**Figure 1: Please state the source of the DEM in this map. Also, please indicate in the figure caption what are the black rectangles, and why some parts of the river appear blue while others are gray. Is it possible to add the Xingu River to the map?**
Reply: the DEM used in the map is from MERIT DEM. We included this infomation as well as the nature of the black rectangles (limits of nautical charts), of the blue lines and of the gray polygon in the revised caption (see Fig. 1).

**L40: Please consider replacing the word "with" with the word "extending".**
Reply: Ok, done.

**L58: "anthropic" – should this be "anthropogenic"?**
Reply: Yes, corrected.

**Figure 3: Is the yellow area in panel a always wet? If so, please indicate it in the figure's caption or somewhere along the river.**
Reply: Yes we confirm. For clarity we modifed the figure in the revised version (now Figure 4), to make this part of same colour as the 100% flood frequency.

**L144: The reference to Fassoni-Andrade et al, 2020 is in place, however, a similar methodology to map water bodies was also published in an earlier study (Armon et al., 2020).**
Reply: The reviewer is right. We included it accordingly (line 148).

**L253: "top-to-raster" – should this be "topo-to-raster"?**
Reply: Sorry for that. We corrected it (line 271).

**References**
Armon, M., Dente, E., Shmilovitz, Y., Mushkin, A., Cohen, T. J., Morin, E. and Enzel, Y.: Determining bathymetry of shallow and ephemeral desert lakes using satellite imagery and altimetry, Geophys. Res. Lett., n/a(n/a), e2020GL087367, doi:10.1029/2020GL087367, 2020.

**Reviewer #2 (Dr. Thierry Schmitt)**

**General comments**
**As reviewer, I recognize the quality of the work undertaken to realize a topo-bathymetric DEM in such a complex environment. I am particularly attentive to actions related to the tedious work of digitizing charts, plus all the detailed work done to properly take into account the different vertical datums and above all the innovative way of getting elevations from the flood exceedance method. The reviewer appreciate that you illustrate the inter-relation between these datums as it brings forward to the mind of the reader this issue in terms of vertical accuracy of your data product.**
Reply: We are thankful for your appreciation and for your thorough and constructive review.

**Specific comments**
**Line 113:As many planar interpolation tools are existing to grid point cloud data, could you please further detail why you have selected the topo-to-raster algorithm (general principle, pros and cons of this algorithm).**
Reply: In fact, many algorithms could be used for interpolating the points on the river, such as kriging. Each algorithm has pros and cons and the result is not very different if the primary data have a relatively good resolution compared to the typical scale of the bathymetric features targeted in the mapping, as it is here. In this case, the largest source of uncertainty is the accuracy of the depth estimates provided in the nautical charts rather than the choice of the interpolation method. We chose the topo-to-raster algorithm because it is a method widely used in watersheds mapping, as it preserves the drainage network. The method is meant to clean up spurious drains on the generated surface and forces a connected drainage structure.

**Section 2.2: We suggest a schema to better describe your methodology and make it easy for the reader to understand it.**
Reply: Thanks for the suggestion. We have added a flowchart of the methodology and data used in the study (see Figure 3 of the revised manuscript). We believe that this schematic facilitates the interpretation and reproduction of the method.

**Figure 3.a I believe the color ramp is not fully appropriated to what you want to describe. Indeed you are using a divergent color map centered close to 50%. It would be preferable to use a sequentially progressing color map (like blue to red, or light yellow to brown, without going through white).**
Reply: We agree. We have changed the color ramp with a classical rainbow palette, in the revised figure (now Figure 4).

**Few comments on the analysis of this map: I do not understand the strong boundary between dark blue and yellow (SSW of the map). Also, it is my understanding that flood exceedance should be high close to the border of the river (banks, coast) and small far away from it (where the altitude is higher). It does appear to be the opposite from your map. Please comment or correct. If this a poor interpretation from me, please help me (and other readers) not get confused and/or better understand your methodology and how to interpret this map.**
Reply: We unified the color code throughout the domain to have the south-western part of the domain of same shade as the rest, for always-flooded areas (now consistently in dark red). The reviewer is correct about the flood frequency (FF): higher values near the river (FF=100% in river) and lower at high altitude. In the previous map, the SRTM DEM was on a black and white background and we believe this made it difficult to interpret the map. Also,

the strong boundary between dark blue and yellow in the previous map represents regions of the levees near the river that have lower FF than the floodplain lakes. We removed the SRTM and changed the color ramp, and now the interpretation is clearer (Fig 4). Note that the floodplain has lower FF values while the high altitudes are never flooded (in white on the map).

**Section 2.3.3. This section seems to indicate that a 2D representation of the WSE would be welcome.**
Reply: 2D representation (x and y) of the WSE along the A-F sections, when available, would not bear much relevance because the mean water slope along the section is insignificant. Besides, 2D representation in time and space (x) is not feasible as such information is not available. It would be necessary to use a hydrodynamic model that resolves the tidal propagation sufficiently well, which is still to be developed over the region. Therefore, only the 1d representation of the water level was presented (Fig 2).

**GEBCO: GEBCO is mainly originating from Hydrographic Offices data. These organizations tend to provide bathymetric data relative to local "Chart Datum" which can be roughly estimated to be equivalent to the Lowest Astronomical Tide. While in deep ocean (main objective of this DTM) Lowest Astronomical Tide (LAT) and Mean Sea Level (MSL) can be safely assimilated to be convergent, it is not the case in coastal waters. To my knowledge, no vertical shifting consideration from CD to MSL has been taken into account in the overall GEBCO production. This should be considered as a serious limitation having major consequences on the vertical precision of the bathymetric DTM.**
Reply: We agree. Indeed, as stated in GEBCO portal (www.gebco.net/data_and_products/gridded_bathymetry_data/, last accessed 23/5/2021), "GEBCO's global elevation models are generated by the assimilation of heterogeneous data types assuming all of them to be referred to mean sea level. However, in some shallow water areas, the grids include data from sources having a vertical datum other than mean sea level". To express this limitation, we added the following sentence in Section 5.2 (lines 403-409): "*Inherently, our product relies on GEBCO digital terrain model in the open-ocean region. As such, we are subject to the same sources of error as everywhere else in the world ocean, related to the poor knowledge of the vertical datum of some of the primary data used in GEBCO composite product*."

**Another important point to consider with GEBCO is that it is a composite DEM with multiple sources of different intrinsic horizontal and vertical resolution. Locally, using the TID (accompanying grid) we can see that bathymetric information in the GEBCO grid are originating from "interpolated based on a computer algorithm" but we do not have any more details on the underlying data and their characteristics (density, vertical and horizontal accuracy). Hence, it is not fully advised to rely on the transition between land and sea from GEBCO.**
Reply: We agree. Still in the absence of any reliable alternate product in the coastal oceanic part of our region of interest, we are left with this source of uncertainty. To some extent, this potential problem is attenuated by our remote-sensing-based approach implemented throughout the intertidal domain. For clarity we incorporated the following sentence in Section 5.2 (lines 409-410):
"*Our product is also potentially impacted by the inhomogeneity of the quality of GEBCO digital terrain model, in particular in the near-shore oceanic regions*".

**Figure 4: Here again I believe the color-ramp is not adequately chosen (gray at 4m can be easily confused with cyan at -1m) prefer here also a sequential color-ramp.**
Reply: We agree. We have changed the color ramp to blue-red sequential (see Fig. 5 of the revised manuscript).

**Figure 6. Please indicate the vertical reference in the legend. Also note that as Bed elevation and Terrain elevation should have a complementary color bar while they seem to intersect between 0 and -1m**
Reply: We agree. We have included the vertical reference (EGM08) in the legend (Fig. 7 of the revised manuscript). We have also included the different mapping domains. We have changed the color bar of the unified mapping.

**Conclusion/Caveat: I would have preferred a more detailed discussion section rather than a "caveat section" in the Conclusion. However, I recognize the rigorous and "ethical" state of mind of the authors in stating the limitations of their data product.**
Reply: Thanks for the suggestion. We believe the text of the manuscript, in its present version, is already quite long compared to the typical articles published in ESSD. For this reason, we would prefer to stick to a concise concluding section (Section 5).

**May I also suggest to the authors to:**
**- undertake a simple slope or shading (in different orientation) which may highlight some artefacts or poor discontinuities in their model**
Reply: Sorry, we did not understand what the reviewer means. We posted a question on the editorial web page to understand better his concern. We are willing to do the needful.

**- provide a mask indicating where the is the limit of your DEM, with the level of accuracy you describe in your paper.**
**Moreover, I would suggest providing an associated grid describing the origin of each nodes of your DEM (like the TID concept for the GEBCO grid).**
Reply: Thanks for the suggestion. We have included in our Mendeley repository (http://dx.doi.org/10.17632/3g6b5ynrdb.2) a shapefile with the boundaries of each domain of our DEM (domain_boundary.shp): 1) Bathymetry (nautical charts); 2) Topography of periodically flooded; 3) MERIT DEM; 4) GEBCO v2020 and 5) Interpolated areas.
We also included in this file the RMSE of bathymetry and topography.

**May I also suggest the following reading:**
**[1] P. Weatherall et al., "A new digital bathymetric model of the world's oceans," Earth Sp. Sci., vol. 2, no. 8, pp. 331–345, Aug. 2015.**
**[1] C. J. Amante and B. W. Eakins, "Accuracy of Interpolated Bathymetry in Digital Elevation Models," J. Coast. Res., vol. 76, pp. 123–133, 2016.**
Reply: The authors thank the Reviewer for these relevant reading suggestions. We included these two references in the revised manuscript (lines 409-411).

**Reviewer #3 (Dr. Panagiotis Agrafiotis)**

**I read the article with great interest and for sure it contributes to the domain. Data are quite difficulte to capture and releasing them freely available and online is of high importance. Some questions and remarks can be found in the attached PDF file. I am suggesting major revisions since i believe that authors should clarify some parts of the text under question and revise the text for english language.**
Reply: We are thankful for your evaluation and for your detailed comments.

**-l. 22: "accuracy": Planimetric? Vertical? I belive that you mean vertical (depth accuracy).**
Reply: Yes, we mean vertical accuracy. We corrected the sentence accordingly (line 22).

**-l. 22: " 8.4 m": Use half non-braking indent between the number and the units**
Reply: We modifed the space accordingly. Thanks.

**-l. 27: " ocean. However, there does not exist, up to now": Suggestion: However, up to now, there is not...**
Reply: Done (line 27).

**-l. 63": " floodplains. These various domains being inherently different in nature, our methodological approach is twofold": Rephrase.**
Reply: We have removed this phrase (line 63).

**-l. 84: " 1. The primary bathymetric surveys utilized in these charts (...) noted SYZ).": What's the expected accuracy of this method? Can you estimate the error propagation between those methods and you dataset? Are these data categorized using some standards i.e. IHOs?"**
Reply: To the best of our knowlege, these references were decided independently of IHO. The expected accuracy is also hard to know, in the absence of any metadata other than the nautical charts themselves. To the best of our knowledge, this information is unavailable from the data provider (Brazilian Navy). As a result it is, unfortunately, virtually impossible for us to attempt an assessment of the error specifically originating from these references and propagated to our dataset.

**-l. 125: Figure 2: Is this for the same area right?**
Reply: The two regions are immediately adjacent, as can be seen from the continuity of the isobaths from one to the other. For clarity we stated it in the figure caption (line 133).

**-l. 220: "We assume that target information from the water represents only the water surface elevation.": Haven't you done any test to prove this assumption?**
Reply: This point was also raised by Reviewer #1. This assumption is indeed hard to prove, in the absence of systematic data of the water turbidity in the Amazon, to the best of our knowledge. Still, we believe it is a reasonable statement, as Parrish et al. (2019; reference cited in the preceding sentence of the manuscript) concluded that ATLAS instrument is basically blind beyond 1 Secchi of water depth, which typically amounts to less than 30 cm in the Amazon (Nakajima et al. 2017, doi: 10.7717/peerj.3308).

**-l.225: "only ATL08 points from the October-December seasons of 2018 and 2019 were considered": This applies to all the manuscript: How this variability in season/years etc**

**affects your data? Have you checked what's the repeatability of the depths from data acquired in different years?**

Reply: Our bathymetry and topography data are represented by the terrain elevation relative to a vertical reference (EGM08) - not relative to water depth. Seasonal and interannual variability in water depth does not affect these data since bathymetry was estimated from a calculated reference water level and topography does not depend on this information for the estimate. However, the floodplain area mapped depends on the variability of the flood extent. We consider 4 years of these mappings (2015-2018) that are representative of excess flood (2015), drought year (2016), and more normal years (2017, 2018). We have added the following sentence (lines 126-128):

"*This period of 48 months was found to be a good compromise between a short enough period to ensure the dataset is recent enough, and a long enough period capable of capturing the bulk of the flooding statistics*".

**-l. 253: " Since GEBCO data has integer values at intervals of 1 meter, the top-to-raster interpolation was used considering the 1 m isolines to generate data consistent with float values wiping out staircases artefacts.": Many interpolations have been used in your methods and it is normal. However, can you estimate how accurate is this?**

Reply: Unfortunately, we were not able to estimate the accuracy of the interpolation, in the absence of independent bathymetric data over the oceanic part of our domain.

**-l.257: " However, since MERIT DEM represents the topography of 2010 and some areas in the coastal region may have been eroded or accreted between 2010 and the 2015-2017 period considered in the flood frequency mapping, a procedure was implemented to correct this issue considering three types of regions" : Indeed, it is necessary.**

Reply: We agree.

**-l. 277: " data. The estimated topography yields an RMSE of 1.15 m, a bias of -0.78 m, and a Pearson correlation coefficient ($r$ ) of 0.52 (number of data, n = 612) compared to GEA/EB DTM.": Vertical? Planimetric? I assume you mean vertical. Please clarify in the text. What's the accuracy in the 95.4% confidence level?"**

Reply: Yes, we mean vertical accuracy indeed. To make it clear, we have changed "*estimated topography*" to "*estimated elevation*" in (line 296). We understand that vertical accuracy in 95% confidence level (sigma = 1.71 m) is an adequate metric when bias is removed. Since there was no removal of bias, we included the standard deviation of bias (line 297).

**-l. 281: " Considering the ICESat-2 data, the estimated topography was also well represented in the riverbanks/floodplain and coastal area with $r$ of 0.8 and 0.8, RMSE of 1.5 m and 1.8, and a bias of 0.9 m and -1.5, respectively.": Same as above.**

Reply: We included the information ("*terrain elevation*") in line 300.

**- L. 290: Figure 4: In (e) there is a bending on the data at about 50% indicating a non-linear relation between the data under 50% and the data above. Why is that? A 2nd degree polynomial function should fit better.**

Reply: We do not fit any functions to the data, our goal with the linear regression is just to visually highlight the trend of the data in the figure. A possible explanation for this trend, also mentioned by Reviewer #1, is provided at the end of page 2 of the present document.

**-L. 296: " RMSE of 8.4 m": Vertical**

Reply: Included (line 316).

**-L. 296: " average bias of 4.3 m " : Please provide more statistics. Std., mean value, etc.**
Reply: We included the standard deviation values (line 316).

**-L. 305: " Sections B, C, and D, seems to have undergone the most significant bed change. In general, the errors are larger (e.g., RMSE of 16 m in Section C)": Why is that?**
Reply: The likely reason for largest errors oberved in this sub-region is the high temporal variability in the riverbed bathymetry. We have added this information in the text (line 333-336):
*These are areas of intense sediment transport with erosion and/or deposition of sand and high variability in the riverbed morphology (Vital et al., 1998). Marked seasonal changes in the riverbed were reported due to extreme net erosion, such as the modification of a channel from a wavy bedform during rising discharge (January of 1994) to a flat floor during low discharge (November of 1994) with the reduction of up to 7 m in the channel depth (Vital et al., 1998). Therefore, an accurate riverbed mapping of this region remains challenging.*

**-L. 349: "accuracy": vertical. Have checked the Hz accuracy?**
Reply: We did not calculate the horizontal accuracy of our mapping and it would be quite hard to do, as the dataset is very diverse in nature. This error would typically amount to a few meters, which is much smaller than the error related to topographic change due to erosion and deposition of sediments over the four years considered.

**-L. 368: " Our product's main limitation lies in the long timespan of our raw bathymetry data collection": correct.**
Reply: We agree.

**Reviewer #4 (Dr. Marco Ligi)**

**General comments**
**This is an original and very good technical paper that addresses data collection, compilation and merging of space born survey data, river depth and global topography and bathymetry data to obtain a comprehensive high-resolution grid of the topography and bathymetry of the Amazon estuary. This dataset constitutes the basic knowledge essential to understand the complex dynamics, morphology and related ecological processes of the entire Amazonian estuary environment, the largest in the world. In general, the manuscript is well organized and well written. The data presented to support the authors' main goals are compelling. The title well represents the content of the paper; the methods and conclusions adequately support the dataset presented. It might be meaningful and appropriate after a minor review for the journal "Earth System Science Data". All parts of the manuscript and illustrations are needed to show the results and to understand the main points.**
Reply: We are thankful for your feedback and for your detailed review.

**Specific comments:**
**My comments/suggestions are minor and mainly aimed at better clarifying to the reader some aspects of the techniques/methods used:**

**Vertical reference level (WSE): the vertical reference levels (WS90 and SYZ) were inferred over the entire study area after the curvilinear interpolation of WSE calculated in seven gauge stations along the Amazon River (lines 106:110) and are shown in Fig. 2c. Which polynomial interpolator was used and on how many points did you use for it (Newton, Lagrange, etc.)? Looking at the figure it seems that the values â␣‹â␣‹between the stations are linearly interpolated. Furthermore, in the caption of Fig. 2c it should be reported what the dashed blue and black lines represent.**
Reply: The levels between the stations have been linearly interpolated between the 8 stations, as in figure 2c. We have added this information in lines 113-114: *"The linear interpolation considered successive points along the river spaced by 30 m and represented by the two blue lines in Fig. 1."*
We also included the explanation of the blue and black lines in the caption of Figure 2c.

**Ground truth (in situ surveys of the river bathymetry): six cross-sections were acquired for data validation along the river (lines 233:236). River water depths were obtained from an Acoustic Doppler Current Profiler (ACDP). This is not a common technique for offshore bathymetric surveys, thus may be interesting to provide the reader with further details on this technique, i.e., what are the advantages relative to conventional echosounders and/or multibeam systems for using ACDP in rivers? Different parameters affect water column velocity in a river, how about the calibration of the system?**
Reply: We acknowledge that using ADCP measurements to infer the river bathymetry is not common practice, and that in general it is more straightforward to use conventional echosounder transects. However, given that this dataset was the sole available to us for an independent validation of our product, we had no other option but to consider it. The surveys were carried out by the Brazilian Geological Service, which, following the standard international practices such as the ones used by USGS, uses acoustic doppler equipment to determine the flow rates of its fluviometric stations. The Brazilian Geological Service routinely uses the RiverRay ADCP device (from Teledyne RDI), which is known to have a

depth estimation with less than 1% error in this kind of environment.

**It is not clear how river bed data from in situ surveys have been corrected for tide amplitudes away from gauge stations, in particular for cross-sections B, C and D. Tide correction depends on time and space. How did you propagate over time values recorded on February 6th, 2007 at the Porto de Moz station located on the South Channel in these cross-section points? Have you assumed the same phase of M2 and S2 tidal components at points B, C and D as that observed at the measured station (only amplitude variations of M2 and S2 may be estimated from the WSE slope in fig. 2c)?**
Reply: We acknowledge the text was not clear. For sections B, C and D, we considered the WSE at Porto de Moz observed on the day of the survey. Besides, to correct the difference in water level (Dh) between the station and the sections, we consider the slope of the water (Fig. 2c) and the distance between the station and the sections. That is: Dh = slope x distance. We acknowledge our approach neglects the tidal variations of the water level along the time span of the shipborne surveys but in the absence of knowledge of the tidal water level there during the course of each survey, we have no better option. We remind that the tidal variability is weak over this region (peak-to-peak variability of order 50 cm, see Kosuth et al. 2009). We have corrected the text (line 252-255):
*"In sections, B, C, and D, the WSE at Porto de Moz station on the day of the survey was used and corrected for each section considering the water surface declivity obtained by the WSE estimated along the river (Fig. 2c) and the distance between station and section, i.e., WSE at section = WSE at Porto de Moz + WSE slope×distance."*

**The vertical reference level (WS90 or SYZ) should not depend from time at a given point, what does it mean that the WSE was measured every 15 minutes at Porto de Santana station? (line 239).**
Reply: Since sections E and F are extensive (~12 km) and as this region is subject to marked tidal variation, the water level varied throughout the survey (~50 cm). This means that for each point the WSE at Porto de Santana at the time of the survey was considered. We acknowledge this information was not clear hence we have modified the text (lines 255-258):
*" the WSE at Porto de Santana station on 5 June of 2008 was considered in sections E and F. Since the water level varied by ~50 cm during the time span of these sections, described in Callède et al. (2010), the high-frequency WSE at each point of the sections was considered."*

**Ancillary database (GEBCO2020): GEBCO is a bathymetric compilation from different data sources: multibeam, single beam and gravity predicted bathymetry. Lines 254:256, "a was applied to reduce in situ multibeam sounding swath edges", this sentence is unclear. There are only few multibeam lines in GEBCO from the study area.**
Reply: GEBCO has indeed regions of artificially high slope due to the merge with multibeam sounding data. One of them is displayed in the following figure as an example. Over our domain of interest, several such multibeam swaths exist in GEBCO data (Weatherall et al., 2015; reference cited in the revised manuscript). These artefacts are problematic as they yield very large values of the gradient of the bathymetry along their edges. This quantity of the bathymetric gradient being an essential ingredient of the hydrodynamical modellers, who are a central taregt among the future users of our dataset, we decided to dedicate specific effort to curb this problem. To reduce these errors, we apply a low-pass filter to bathymetry, with different cut-off parameters in the off-shore (offshoreward of the -200m isobath) and on-shore region (shoreward of it) (see the figure in the next comment). We have modified the text as follows (lines 273-275):
*"Besides, a low-pass filter with a 9 × 9 points and 19 × 19 points window moving average,*

*i.e., 4.5 km × 4.5km and 9.5 km × 9.5 km respectively, was used in the region, respectively, above (shoreward) and below (off-shoreward) the -200 m isobath to reduce the noise caused by in situ multibeam sounding swaths edges."*

[Figure]

**The main problem with the final merged 30 m grid is the evident noise related to gridding in the offshore part of the study region. When hill-shading is applied to the grid, this noise is the most noticeable feature. Probably, the final grid in the offshore area may be improved applying a smooth interpolation method to GEBCO data in order to reduce grid size from 450 m to 30 m (to avoid aliasing) before merging all the data using the topo-to-raster method.**

Reply: We acknowledge the residual noise level in GEBCO product can be an issue. The figure below exemplifies an offshore region with the original GEBCO (~ 460m) and the final grid (30m), both with the hillshading effect. Note that noise related to gridding is present in both data. Therefore, we believe that neither the change of spatial resolution to 30m nor topo-to-raster interpolation did not produce this effect.

Indeed, the hillshading shows some terrain slopes, but they are very small in relation to the other corrected errors and also present in the original GEBCO. We can not anticipate the variety of uses that will be made with our dataset by the various scientific communities we target with our product. Each of them will certainly require a tailor-made smoothing procedure to our dataset, if any. Thus, we chose not to apply any further filtering on the final grid.

[Figure]

GEBCO 460 m

Final grid 30m
(low pass filter -9.5 × 9.5 km)

**The several datasets collected were referred to different geodetic datums. How do you shift all the data to EGM08 reference system.**

Reply: The bathymetry and topography we created in the estuarine part of our domain were referenced from the water level in relation to EGM08 across the domain, from the vertically levelled tide gauge stations we utilized, as explained in Section 2.4 (line 268). The MERIT DEM is available in EGM96 so we converted to EGM08 by simply adding the difference between EGM96 and EGM08 models; this was stated in section 2.2 (line 136). As for GEBCO, as also raised by Reviewer #2, it doesn't have any specific vertical reference, so we considered the values to be above the geoid, as is the common practice (line 269).